



# High resolution downscaling of CMIP6 Earth System and Global Climate Models using deep learning for Iberia

Pedro M. M. Soares[1], Frederico Johannsen[1], Daniela C. A. Lima[1], Gil Lemos[1], Virgílio Bento[1], Angelina Bushenkova[1]

[1]Instituto Dom Luiz, IDL, Faculty of Sciences, University of Lisbon, 1749-016 Lisbon, Portugal

*Correspondence to*: Frederico Johannsen (jfjohannsen@ciencias.ulisboa.pt)

**Abstract.** Deep learning (DL) methods have recently garnered attention from the climate change community, as an innovative approach for downscaling climate variables from Earth System and Global Climate Models (ESGCMs) with horizontal resolutions still too coarse to represent regional-to-local-scale phenomena. In the context of the Coupled Model
Intercomparison Project phase 6 (CMIP6), ESGCMs simulations were conducted for the Sixth Assessment Report (AR6) of the Intergovernmental Panel on Climate Change (IPCC), at resolutions ranging from 0.70º to 3.75º. Here, four Convolutional Neural Network (CNN) architectures were evaluated for their ability to downscale, to a resolution of 0.1º, seven CMIP6 ESGCMs over the Iberian Peninsula - a known climate change hotspot, due to its increased vulnerability to projected future warming and drying conditions. The study is divided into three stages: (1) evaluating the performance of the four CNN
architectures in predicting mean, minimum, and maximum temperatures, as well as daily precipitation, trained using ERA5 data, and compared with the Iberia01 observational dataset; (2) downscaling the CMIP6 ESGCMs using the trained CNN architectures and further evaluating the ensemble against Iberia01; and (3) constructing a multi-model ensemble of CNN-based downscaled projections for temperature and precipitation over the Iberian Peninsula at 0.1º resolution throughout the 21st century, under four Shared Socioeconomic Pathway (SSP) scenarios. Upon validation and satisfactory performance
evaluation, the DL downscaled projections demonstrate overall agreement with the CMIP6 ESGCM ensemble in terms of temperature and precipitation projections. Moreover, the advantages of using a high-resolution DL downscaled ensemble of ESGCM climate projections are evident, offering substantial added value in representing regional climate change over Iberia. Notably, a clear warming trend is observed, consistent with previous studies in this area, with projected temperature increases ranging from 2ºC to 6ºC depending on the climate scenario. Regarding precipitation, robust projected decreases are
observed in western and southwestern Iberia, particularly after 2040. These results may offer a new tool for providing regional climate change information for adaptation strategies based on CMIP6 ESGCMs prior to the next phase of the European branch from the Coordinated Regional Climate Downscaling Experiment (EURO-CORDEX) experiments.



# 1 Introduction

The Sixth Assessment Report (AR6) of the Intergovernmental Panel on Climate Change (IPCC) was released in August
2021, dramatically calling for urgent action to reduce global greenhouse gas emissions (GGE) due to the scale of projected
changes for the climate system, from the mean state to extremes (IPCC, 2021). The extensive results presented are based on
the Coupled Model Intercomparison Project phase 6 (CMIP6) simulations, which were performed using Earth System and
Global Climate Models (ESGCMs), and included runs with spatial resolution in the range of 0.70º to 3.75º. The IPCC report
projects worrying changes in what concerts to global-scale extreme events, such as significant increases in the frequency and
intensity of heatwaves, droughts and extreme precipitation. Although based on global simulations, the AR6 offered a
regional view of those changes being especially intense, in some climate change hotspots, like the Mediterranean region
(Turco et al., 2015; Cos et al., 2022; Lionello and Scarascia, 2018).

It is widely accepted that most resolutions used by ESGCMs are still too coarse to represent many regional to local scale
processes that define the local climate (Randall et al., 2007; Soares et al., 2012; Rummukainen, 2016). Such disadvantage
fosters the need for downscaling methods, at higher resolution, which often provide regional to local fine scale information,
crucial for impact and adaptation studies. There is a plethora of downscaling methods, including dynamical ones using
Regional Climate Models (RCMs), statistical ones using Statistical Downscaling Methods (SDMs) and, recently, an
umbrella group of the latter, designated as Artificial Intelligence (AI) approaches, which include Machine Learning (ML)
and Deep Learning (DL) methods.

RCMs are forced at the boundaries by ESGCMs (Dickinson et al., 1989; Giorgi and Bates, 1989; Giorgi and Mearns, 1991;
McGregor, 1997; Christensen et al., 2007), using higher resolutions (~10km) in limited area domains, which improve
significantly the description of regional to local climates (Giorgi and Mearns, 1999; Laprise, 2008; Rummukainen, 2010,
2016; Feser et al., 2011; Soares et al., 2012, 2017a,b; Rios-Entenza et al., 2014; Giorgi et al., 2016; Lucas-Picher et al.,
2017; Cardoso et al., 2019). Nevertheless, considering local and, especially, sub-daily climate features, RCMs still present
limitations in capturing sub-grid processes such as convection (Prein et al., 2013). In order to bridge this gap, RCMs are
running at very high resolutions, usually described as convective permitting resolutions (approximately 1 km), where deep
convection is explicitly resolved by the grid mesh at grid spacing below 3 km (Prein et al., 2015; Coppola et al., 2020;
Pichelli et al., 2021; Soares et al., 2022a).

SDMs are based on the establishment of empirical relationships between large-scale atmospheric predictors and local
observed predictands describing local climate (Wilby and Wigley, 1997; Fowler et al., 2007; Nikulin et al., 2018; Hertig et
al., 2019; Maraun et al., 2019; Gutierrez et al., 2019; Rössler et al., 2019; Soares et al., 2019; Widmann et al., 2019).
Subsequently, projections of future regional to local climate variables are determined from future large-scale atmospheric
conditions. SDMs include model output statistics and perfect prognosis approaches (Maraun et al., 2010; 2017). However,
when compared to dynamical downscaling, the model formulation of SDMs lack physical constraints and, in general, do not
ensure a full multivariate consistency (Le Roux et al., 2018). Since SDMs use observations for training, they are able to



overcome the systematic biases often displayed by RCMs. Additionally, since SDMs are not computationally demanding, it avoids the need for large computational infrastructures.

There is a continuous improvement in SDMs, and new AI approaches are being proposed for climate applications, with Deep Learning (DL) being one of the most promising ones. DL is a subdomain of Machine Learning (ML) which, in turn, is a subdomain of AI. In ML, the models train by themselves, learning the optimal value of their parameters automatically. Since parameter tuning is based on the input data fed to the model, the model is able to make predictions when forced by new data (see Alzubi et al. (2018) for an overview of ML). Unlike "shallow" learning models (e.g., Random Forests, Support Vector Machines), DL models learn non-linear relationships between data due to their "deep" layered structure. DL has become a common approach in research over the past decade (Schmidhuber, 2015), including in Earth Sciences in the last few years (Reichstein et al., 2019), thanks to advances in computational power and data availability. For example, the European Centre for Medium-Range Forecasts (ECMWF) features DL as the main showcase in its Destination Earth project (Bauer et al., 2021) that will attempt to create Digital Twins of the Earth System in the next decade.

The most common DL model type is the Artificial Neural Network (ANN), an attempt to design an artificial analogous to the biological neural networks that exist in the human brain. One of the most used types of ANNs are the Convolutional Neural Networks (CNNs). These models are widely used in the field of Computer Vision, as they extract information and identify objects in images (LeCun and Bengio, 1995). However, CNNs' value is not restricted to Computer Vision, as CNNs have been used in other research areas, including in Earth Sciences, for example in model parameterization (Chantry et al., 2021a) and ensemble postprocessing (Rasp and Lerch, 2018), showing promising results. Climate downscaling is another promising area benefitting from the implementation of CNNs. There have been early attempts of downscaling using simple ANN structures, but the results were not compelling enough due to limited input data, computational resources and scarcer observations (e.g. Wilby et al., 1998; Trigo and Palutikof, 1999). Recent studies have shown more favourable results, equalling and even surpassing classic SDMs (e.g. Baño-Medina et al., 2020; Hernanz et al., 2022; Baño-Medina et al., 2022). Recently, and for the first time, Baño-Medina et al. (2022) was able to downscale climate projections with the aid of DL for precipitation and temperature, based on a set of GCMs from CMIP phase 5 (CMIP5). These authors showed that DL reduced the biases in the historical period when compared to an ensemble of RCMs with 0.44° resolution, from EURO-CORDEX (European branch from the Coordinated Regional Climate Downscaling Experiment). In addition, the resulting climate change signals have similar spatial patterns to those obtained from the RCMs, and when looking at the uncertainty, the DL preserves the uncertainty of the climate change signal for temperature and reduces for precipitation.

Despite their promising results, DL methods are perceived with precaution in the scientific community due to their black-box nature. DL models usually have thousands (if not millions) of trainable parameters that hinder a physically-based explanation for the quality of their results. There have been attempts to improve the understanding of models' reasoning (e.g. Carter et al., 2018), attempts at building an overall framework for DL studies in Earth Sciences, including weather/climate modelling and postprocessing, and to generate consistent intercomparable studies (Reichstein et al., 2019; Chantry et al., 2021b; Haupt et al., 2021). As a result, the first benchmark dataset for data-driven weather forecasting has been created



(Rasp et al., 2020). DL also presents other general limitations, including the need for hardware (GPUs accelerate the model
     training while the more common CPUs can be computationally costly; Chantry et al., 2021a), and limitations specific to the
     climate research field: for example, lack of explicit physics in the DL models, and the need to split the data in a way that
     includes long-term patterns and trends (e.g. ENSO and global warming) in both training and test phases for long-term
     datasets (Schultz et al., 2021).

The Iberian Peninsula, within the Mediterranean basin, is a known climate change "hotspot" (Planton et al., 2012;
     Diffenbaugh et al., 2012; Turco et al., 2015; Russo et al., 2019; Cos et al., 2022) due to its high vulnerability to warming and
     drying conditions (Argüeso et al., 2012; Cardoso et al., 2019; Soares et al., 2017; Lima et al., 2023a,b; Soares et al., 2022b),
     leading to strong impacts on the occurrence of extreme events, such as droughts, heatwaves, and wildfires (Hoerling et al.,
     2012; Bento et al., 2022; Bento et al., 2023; Soares et al., 2023). Future projections point to a warming trend stronger during

summer and autumn seasons, and during daytime, resulting in an amplification of the daily and annual temperatures
     (Cardoso et al., 2019; Lima et al., 2023a). Also, it is projected a significant reduction of the mean precipitation along the
     entire year (Argüeso et al., 2012; Lima et al., 2023a; Soares et al., 2017). Aligned with the projected warming and drying
     trends, the occurrence of hot and dry extreme events is expected to become more frequent, intense, and longer (Hoerling et
     al., 2012; Lima et al., 2023b), which may have significant impact on human and natural sectors, such as agriculture (Bento et

al., 2021), forests (Palma et al., 2015; Palma et al., 2018), coastal areas (Pereira et al., 2013), water resources (Soares et al.,
     2022b).

     The most consistent and widely used high-resolution climate change dataset for Iberia remains the EURO-CORDEX and
     CORDEX-Core runs (Jacob et al., 2014; Jacob et al., 2020). These regional climate simulations were forced by the previous
     CMIP5 global climate simulations and are becoming less useful after the recent release of the CMIP6 results forced by the

Shared Socio Economic (SSPs) - Representative Concentration Pathways (RCPs) greenhouse gas emissions scenarios. At the
     present date, the new EURO-CORDEX simulations protocol, to be forced by CMIP6 runs, is being finished, and widespread
     availability of new simulations and results for the scientific community and society is not expected before one-two years'
     time. Additionally, the building of new multi-model and multi-approach ensembles is highly beneficial to assess robustness
     and uncertainty of future climate projections (Lima et al., 2023a). The increasing need for exploring and updating regional

climate information for Iberia requires and benefits from the use of other approaches to downscale the current CMIP6 runs.

     In the present study, a DL methodology based on the work of Baño-Medina et al. (2022) is used and tested to downscale, in a
     consistent manner, the CMIP6 runs at high-resolution for Iberia. A matrix of plausible futures is used to select the CMIP6
     models considered in agreement with the EURO-CORDEX evaluation study (Sobolowski et al., 2023). The DL algorithm is
     trained using ERA5 and compared to the high-resolution regular gridded dataset Iberia01 (Herrera et al., 2019) for the

current climate, covering the period 1979-2014, and then used to downscale future projections in agreement with four SSPs-
     RCPs scenarios: SSP1-RCP2.6, SSP2-RCP4.5, SSP3-RCP7.0 and SSP5-RCP8.5 (O'Neill et al., 2016), for three future
     periods throughout the 21st century: beginning of the century (2015-2040), middle of the century (2041-2070), and end of
     the century (2071-2100). Firstly, different architectures of DL are trained and evaluated for present climate and then multi-





model projections are performed based on a simple "democratic" multi-model ensemble approach. This study is focused on four of the main climate variables and their extremes: minimum, mean, maximum temperatures and precipitation. The main goals of this study are to understand the liability of downscaling CMIP6 GCMs to a much finer spatial resolution using DL, and to take advantage of the growing AI methods to compile information that may be crucial to timely assist mitigation and adaptation plans being developed at the national, regional, and local levels within Iberia.

## 2 Data and Methods

### 2.1 Study area

The Iberian Peninsula (IP) is located in the southwestern tip of Europe (Fig. 1), bordered by the Atlantic Ocean and the Mediterranean Sea. The IP sits in a climate transition zone between the arid and semiarid climates of subtropical regions and the humid temperate climates of northern Europe. Despite having a surface area of less than 0.6 million square kilometres, it shows a diverse climate, with significant regional variations. In fact, while the north and northwest regions are marked by long rainy seasons and temperate summers, the south and southeast are characterized by long and hot summers, as well as by a clear dry season. The interior regions are defined by a continental climate, with hot summers and cold winters. Additionally, local and regional topographic features play a significant role in modulating climate features throughout the IP. Here, the IP domain is considered as the land area between 36ºN and 44ºN and 10ºW and 4ºE (Fig. 1, inside the orange line).

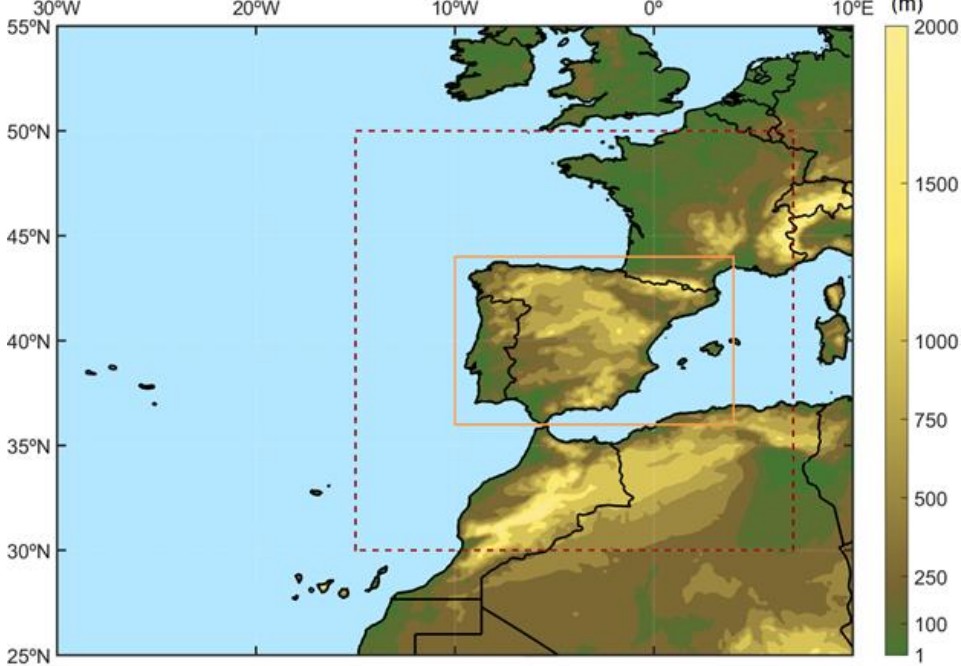

**Figure 1.** Western Europe and Northwestern Africa topography (m) and Earth System and Global Climate Model predictors' (red dashed line) and predictands' (full orange line) domains considered.



## 2.2 ERA5 reanalysis

ERA5 is the latest European Centre for Medium Range Weather Forecasts (ECMWF) reanalysis (Hersbach et al., 2020), produced within the Copernicus Climate Change Service (C3S). ERA5 provides a comprehensive, high-resolution record of
the global atmosphere, land surface, and ocean from 1950 onwards. Benefitting from advanced research and model physics development, outputs are archived at 0.25º x 0.25º horizontal and 1-hourly time resolutions, considering 137 atmospheric levels up to 0.01 hPa. The ECMWF Integrated Forecast System (IFS) Cy41r2, used operationally for forecasting from March to November 2016, is used to produce ERA5. Additional details are available in Hersbach et al. (2020). Here, the period from January 1st, 1979, to December 31st, 2014, is considered. The original ERA5 reanalysis data was interpolated to a 1º x
1º horizontal resolution, using a bilinear interpolation method, to build a common grid to the CMIP6 ESGCMs (section 2.3).

## 2.3 CMIP6 Earth System Global Climate Models

The ESGCMs selected for the current study closely follow the model array built in Sobolowski et al. (2023) for the ongoing CMIP6 dynamical downscaling that is being performed, i.e., the regional climate model simulations of EURO-CORDEX phase II. The authors analysed thoroughly the ability of the CMIP6 ESGCMs to describe the most important large-scale
features that define the European climate, such as the storm-track position, and that span the AR6 IPCC climate sensitivity range. The ESGCMs considered are listed in Table 1; understandably is additionally constrained by the data availability in what concerns to predictors data. The predictor data were extracted for the domain in Fig. 1 (inside the dashed red line), limited by 15ºW - 7ºE; 30ºN - 50ºN, being then interpolated to a common grid at a 1º x 1º resolution using the bilinear interpolation method.

## 2.4 Iberia01 Observational regular gridded dataset

The Iberia01 regular gridded product (hereafter Iberia01) is the highest resolution observational daily dataset including mean, maximum and minimum temperatures and precipitation, covering the full domain of continental Iberia (Herrera et al., 2019). This observational dataset was built using an unprecedented number of ground station observations: 275 for temperatures and 3486 for daily accumulated precipitation, resulting in a high quality regular gridded dataset at 0.1º x 0.1º
horizontal resolution. Iberia01 is commonly used for assessing the performance of ESGCMs (Soares et al., 2022a), RCM results (Herrera et al., 2020; Careto et al., 2022a, 2022b), building of multi-model ensembles for climate change assessments (Soares et al., 2023; Lima et al., 2023a; Lima et al., 2023b) and other studies, such as related to water availability (Soares et al., 2022b) and droughts (Páscoa et al., 2021; Soares et al., 2023). Here, the Iberia01 product is used both to calibrate and evaluate the deep learning approach considering the period 1979-2014 (same as ERA5).




**Table 1.** CMIP6 Earth System Global Climate Models

| ESGCM (CMIP6) | Institute | Reference | Hor. & Vert. Res. Atmosphere | Hor. & Vert. Res. Ocean |
|---|---|---|---|---|
| ACCESS-CM2 | CSIRO / BOM | Bi *et al.* (2012) | 1.25º x 1.875º, L85 | 1.00º x 1.00º, L50 |
| ESM1-2-HR | MPI | Müller *et al.* (2018); Gutjahr *et al.* (2019) | 0.90º x 0.90º, L95 | 0.40º x 0.40º, L40 |
| CM6A-LR | IPSL | Boucher *et al.* (2020) | 1.25º x 2.50º, L79 | 0.5-1.00º x 1.00º, L75 |
| MIROC6 | AORI / NIES / JAMSTEC | Tatebe *et al.* (2019) | 1.40º x 1.40º, L81 | 1.00º x 1.00º, L62 |
| NorESM2-MM | NCC | Seland *et al.* (2020) | 0.90º x 1.25º, L32 | 1.00º x 1.00º, L53 |
| UKESM1-0-LL | UKMO | Sellar *et al.* (2019) | 1.25º x 1.875º, L85 | 1.00º x 1.00º, L75 |
| ESM2-1 | CNRM | Séférian *et al.* (2019) | 1.40º x 1.40º, L91 | 1.00º x 1.00º, L75 |

**2.5 Deep learning methodology**

Convolutional Neural Networks (CNNs) are Deep Learning (DL) model structures specialized in extracting features automatically from geospatial data. The architecture of a CNN model includes convolutional layers that perform feature identification and extraction using filters that apply the mathematical operation of cross-correlation to the data (LeCun and Bengio, 1995; see Fig. 3 of Baño-Medina et al., 2020). The general outline of one epoch, *i.e.*, a full cycle of the training
phase, is as follows:

- The 2D filters in a convolutional layer "scan" the set of predictor variables, computing a set of filter maps based on each filter, highlighting different features/patterns of the original data. These filter maps are then used as input for





the following convolution layer;
- The output of the final convolutional layer is flattened (reshaped to 1D) before being fed to the fully connected (dense) layer that follows;
- The units in a dense layer are connected to every unit in the previous and following layers, allowing the network to learn potential relationships between all units in successive layers. The final dense layer must have the size of the target data in order to generate the predictions;
- The predictions are compared with the observations by calculating the loss according to the loss function defined by the user;
- Finally, the model attempts to lower the loss by the use of the Stochastic Gradient Descent optimization algorithm, tuning the parameters of each model layer according to the direction of the gradient that minimizes the loss the fastest. This process begins in the output layer, computing the gradients on that layer, and backtracks all the way to the first convolutional layer, in what is known as the Back-propagation algorithm.

The model then repeats the training until it reaches a convergence mark defined by the user (usually a set number of epochs after the loss stops decreasing). While the learnable parameters are optimized automatically by the model, there is a set of hyperparameters that is defined by the user, including:
- The maximum number of epochs that the model can run;
- The batch size of observations used to tune the model in each training cycle;
- The learning rate at which the model incorporates new information after each epoch.

The main goal of DL is to achieve generalization, *i.e.*, the ability to make quality predictions when given new, never-before-seen data (extrapolation). Such a feature is particularly important when training DL models for climate studies, due to global warming and other long-term trends. The model structures considered in this study were retrieved from the Baño-Medina et al. (2020) and are described in Table 2. Although all models have similar structures, differing only in small details, they are designed in such a way so that every model is slightly more complex than the previous one. All models comprise:
- Three convolutional layers (the first two layers have 50 and 25 filters each);
- A final dense layer that outputs the predictions;
- The same hyperparameters: batch size = 100 and learning rate = 0.0001.

The differences among the models are the following:
- The third convolutional layer has 1 filter in BMlinear and BM1 and 10 filters in BM10 and BMdense;
- BMdense presents two additional dense layers, both with 50 units, prior to the output layer;
- The activation function in every layer of every model is the Rectified Linear Unit (ReLU), a non-linear function, except in BMlinear, in which the function is linear.

The loss function used for the temperature predictions is based on the mean squared error. The same DL models were used to downscale precipitation, differing however from the temperature ones, so that precipitation models feature a multi-output structure (see Fig. 3 of Baño-Medina et al., 2020). Instead of predicting precipitation directly, the model attempts to obtain three parameters: the shape (alpha) and scale (beta) of the gamma distribution, and the probability of precipitation (p). This is achieved by applying a custom loss function that computes the negative log likelihood of the Bernoulli-gamma distribution (Cannon, 2008), following the methodology presented in Baño-Medina et al. (2020).





**Table 2.** CNN architectures used in this study (adapted from Baño-Medina et al., 2022). The architecture is divided in one input and one output layer and several hidden layers in between. Numbers represent the units in each hidden layer, convolutional layers in bold, dense layers otherwise. The input format is LAT x LON x 15 (5 predictors, times 3 pressure levels) and the output is a 6523 x 1 vector (the number of 0.01° land grid points over Iberia).

| Model | Architecture | Rationale |
|---|---|---|
| **BMlinear** | Input – **50** – **25** – **1** – Output | Using convolutions to perform the downscaling |
| **BM1** | Input – **50** – **25** – **1** – Output | Add non-linearity to the model structure |
| **BM10** | Input – **50** – **25** – **10** – Output | Increase the number of filters in the last convolution layer |
| **BMdense** | Input – **50** – **25** – **10** – 50 – 50 – Output | Deepen the model structure |

## 2.6 Selection of predictors, training, and evaluating

The predictors selected follow the Baño-Medina et al. (2022) study and are included in Table 3. The data was pre-processed before being used to train and evaluate the DL models. The ERA5 variables, used as predictors, were standardized to facilitate the training of the DL models. Any missing data in the CMIP6 ESGCMs was filled and afterwards the dataset was standardized (with the same parameters used for ERA5). The ESGCMs were bias corrected in relation to ERA5 through a simple mean-variance scaling method. The climate change trend was removed in the future scenarios before the bias correction and reintroduced afterwards (Vrac and Ayar, 2017).

Two stages were pursued with the aim of training and evaluating the four architectures. The first stage was to train them using ERA5 predictors (Table 3) considering the 1979–2004 period, validating their performance between 2005 and 2009, and finally testing the architectures for the period 2010–2014. This process was performed to obtain each of the four predictands, namely daily mean temperature (T), daily minimum temperature (Tmin), daily maximum temperature (Tmax), and daily accumulated precipitation (Pr) (Table 3). The results of the DL downscaled predictands from ERA5 were then compared with the Iberia01 reference data. In this case, since the DL used ERA5 reanalysis predictors, the evaluation was performed with daily synchronized climate data. This evaluation, conducted between 2010 and 2014, was based on error metrics such as the bias, the root mean squared error (RMSE), the standard deviation ratio (SDR) and the Perkins skill score (PSS), and the relative operating characteristic skill score (ROCSS).

The mean bias, used for temperature and precipitation is defined as:



$$Bias = \frac{1}{N}\sum_{k=1}^{N}(m_k - o_k) \, , \qquad (1)$$

where $o_k$ and $m_k$ are respectively the observed and modelled time-series, and $N$ is the total number of grid-points.

The root-mean squared error (RMSE), used for temperature and precipitation, is defined as:

$$RMSE = \sqrt{\frac{1}{N}\sum_{k=1}^{N}(m_k - o_k)^2} \, , \qquad (2)$$

The standard deviation ratio, used only for temperature, is expressed as:

$$\sigma_n = \frac{\sigma_m}{\sigma_o} = \frac{\sqrt{\frac{1}{N}\sum_{k=1}^{N}(m_k - \underline{m})^2}}{\sqrt{\frac{1}{N}\sum_{k=1}^{N}(o_k - \underline{o})^2}} \, , \qquad (3)$$

where $\sigma_o$ and $\sigma_m$ are standard deviations of the observed and modelled time-series, respectively, while $\underline{o}$ and $\underline{m}$ represent the respective mean values.

The Perkins skill score (PSS; Perkins et al., 2007) quantifies the model's ability to reproduce the observed probability distribution functions (PDFs):

$$S = 100 \times \sum_{i=1}^{B} min[E_{m,i}, E_{o,i}] \, , \qquad (4)$$

where $E_m$ and $E_o$ are, respectively, the modelled and observed empirical PDFs and $min[E_{m,i}, E_{o,i}]$ is the minimum between the two values. $B$ is the total number of bins used to compute the PDF.

Finally, the relative operating characteristic skill score (ROCSS) is given by:

$$ROCSS = 2 \times \text{Area under the ROC Curve} - 1 \, . \qquad (5)$$

For the extreme values, the 2nd and 98th percentiles of T, the 10th (90th) percentile of Tmin (Tmax), and the 98th percentile of Pr were computed and compared with those from Iberia01 (bias).

The second stage consisted in training the DL architectures with ERA5, this time using the complete 1979–2014 period. These architectures were then used to downscale the CMIP6 ESGCMs for the same period, for each of the four predictands. The resulting DL downscaled ESGCMs, at 0.1° horizontal resolution (ESGCM-DL) are non-synchronized with the Iberia01, and consequently only a statistical comparison was performed. Therefore, Julian years with 365 multi-year daily means were computed for each ESGCM-DL, and for the Iberia01, and a performance evaluation based on the same error metrics as in the

first stage was conducted. Finally, a democratic (simple average) ensemble was built for each architecture, containing seven ESGCM-DL members, and compared to the 1° ESGCM ensemble, the 1° ERA5 reanalysis, and the interpolated 0.1° ERA5 reanalysis.



**Table 3.** ERA5 and CMIP6 predictors and predictands and the respective atmospheric levels.

| Levels | 850 hPa - 700 hPa - 500 hPa | | | | |
|---|---|---|---|---|---|
| **Predictors (daily)** | Temperature (*ta*) | Humidity (*hus*) | Geopotential Height (*zg*) | Zonal wind speed (*ua*) | Meridional wind speed (*va*) |
| **Levels** | Surface | | | | |
| **Predictands (daily)** | Mean temperature (*tas*) | Minimum temperature (*tasmin*) | Maximum temperature (*tasmax*) | Precipitation (*pr*) | |

## 2.7 Future climate projections

The present climate historical period considered here corresponds to 1981-2010. The future climate projections are focused on three periods: 2015-2040 (beginning of the 21st century), 2041-2070 (middle of the 21st century), and 2071-2100 (end of the 21st century), encompassing four CMIP6 SSPs (Rozenberg et al., 2014; O'Neill et al., 2016): SSP1-2.6, SSP2-4.5, SSP3-7.0 and SSP5-8.5. These scenarios range from a strong mitigation level, resulting in low greenhouse gas emissions (GGE), with $CO_2$ emissions cut to net zero around 2075 (SSP1-2.6), to an intermediate trajectory of future GGE, with $CO_2$ emissions maintaining current levels until 2050 and then reducing, but not achieving net zero by 2100 (SSP2-4.5), and finally, two scenarios with increasing GGE: SSP3-7.0 and SSP5-8.5, where the former considers that $CO_2$ emissions double by 2100, and the latter consider an increase of threefold by 2075. In this study, results of future climate projections correspond to anomalies (differences) between the future and the historical climatological values, given by a simple averaged (democratic) multi-model ensemble consisting of all ESGCM-DL outputs. Downscaling using the DL algorithms is performed for each ESGCM considered in this study (Table 1) for the disclosed future periods. As reference, the climate change signal linked to all ESGCMs is also computed. The future ESGCM-DL projected climate of Iberia is analysed in terms of mean climate and extreme values. Anomaly maps for the annual projected changes for Iberia are presented for all variables, where the differences between the 1°-ESGCMs and 0.1° ESGCM-DL projections are highlighted. Boxplots summarizing the projected changes (median, interquartile range, variability) are also presented, for the four predictands.

## 3 Results

### 3.1 Evaluation of DL forced by ERA5

The four DL architectures are trained and validated with ERA5 for the 1979–2004 and 2005–2009 periods, respectively, and finally tested during 2010–2014 against Iberia01 considering minimum, mean, and maximum temperatures, and





precipitation. The performance evaluation metrics are shown in Fig. 2 (T), Fig. 3 (Tmin), Fig. 4 (Tmax) and Fig. 5 (Pr). The comparison between the ERA5 (interpolated to 0.1º horizontal resolution; iERA5 from here on) and Iberia01 is also shown, as reference for all fields (dark grey boxplot).

Considering T (Fig. 2), three main outcomes emerge: (1) all the DL approaches display rather small errors and even slightly improvements in comparison with iERA5, such as concerning RMSE and the PSS; (2) the DL architectures present less variability in accuracy metrics (bias) than iERA5, but in some cases the error distribution of the latter is more closely centered around zero than for the DL outcomes; and, (3) the four architectures present small warm (cold) bias for lower (higher) T extreme values. When considering the total bias, the four architectures show somewhat interchangeable results,

with median values slightly below zero for BMlinear, virtually zero for BM1, and slightly over zero for BM10 and BMdense. The small warm bias found for the 2$^{nd}$ percentile of T is observed both in the DL outcomes and in the iERA5. However, the cold bias found in the 98$^{th}$ percentile of T is only found in the DL outcomes, with values close to zero for the iERA5. Regarding the RMSEs, DL results show lower variability ranges than iERA5, and overall lower median values with increasing DL complexity. The interquartile range for the RMSEs of the DL results encompasses values from 1.25ºC to

1.5ºC. In terms of SDR, in relation to Iberia01, the iERA5 shows the median value closest to 1, however, it also shows the largest interquartile distance and largest variability range (from ~0.90 to ~1.08, compared with ~0.93 to ~1.04 for the DL outcomes). Finally, regarding the PSS, the four architectures show more similarity between distributions of T with Iberia01 than the iERA5. A distinction between the DL outcomes for this error metric is rather unnoticeable.

Regarding Tmin (Fig. 3), the overall results show the following: (1) the DL architectures present better results in comparison

to Iberia01 than the simple interpolation of ERA5, showing lower RMSEs, biases closer to zero and larger PSSs; (2) the four architectures show, to some extent, similar results between them. On a more detailed analysis, median biases presented by the four architectures are all near zero, while the iERA5 shows a median bias larger than 1ºC. Furthermore, when comparing the bias of the architectures to the interpolated ERA5, a lower interquartile range (circa 1ºC) is observable in the first compared to the latter (~2ºC). Additionally, a narrower extreme bias variability range (about 2.5ºC versus about 4ºC,

respectively) is seen. Results for extreme low temperatures (bias p10) are in line with the total bias, nevertheless showing a slight tendency to lower median biases as the complexity of the architecture increases. This is also noticeable in the precision metric, with reduced RMSEs for increasing DL architecture complexity. However, here, BM10 and BMdense show very similar results. All architectures present median RMSEs below 2ºC, being the third quartile of the three more complex ones below this value as well. The maximum RMSE does not surpass 3ºC. On the other hand, the iERA5 shows a median RMSE

slightly above 2ºC, the third quartile close to 3ºC, and a maximum value above 4.5ºC. Similar to the T results in Fig. 2, the median SDR is closer to 1 for the iERA5, nevertheless, the DL architectures show greater variability ranges for Tmin in comparison to T. Among the architectures, BMdense is the one with a standard deviation ratio median closer to 1. Finally, considering the PSSs, it is once again noticeable that the distributions of the downscaled ERA5 using DL and Iberia01 tend to match better with the increase in complexity of the architectures.





When assessing Tmax (Fig. 4), three main results may be highlighted: (1) all the error metrics are improved by the DL methods when compared with iERA5 (2) the DL architectures show much less variability in the biases and RMSEs in comparison to iERA5 (having Iberia01 as reference); and (3) the four architectures show, to some extent, similar results between them. In terms of bias, and considering the four architectures, neither Tmax nor the extremely Tmax show cold or warm biases, being both centered around zero. Conversely, iERA5 shows a cold bias in both cases. Once again, precision

tends to be larger with more complex DL architectures, with BMdense showing lower RMSEs. The Tmax SDR between BM10 and Iberia01 seems to indicate a better agreement than BMlinear, BM1, and BMdense. Nevertheless, the four architectures present SDR values closer than 1 when compared with iERA5. Finally, the matching between the four DL architectures outcomes distributions and Iberia01 is greater than for iERA5. The high-quality DL results for temperatures w.r.t. to iERA5 are rather promising since those variables are assimilated by ERA5.

For Pr, the error metrics from the comparison between the DL downscaled ERA5, iERA5, and Iberia01, are shown in Fig. 5. In this case, between the four architectures, BMdense is the most accurate and precise one, also surpassing iERA5. While the iERA5 shows an overall slightly negative bias (median of -8 mm), the DL outcomes show generally positive values (between 1 and 10 mm). Considering the extreme Pr (98th percentile), all approaches show an underestimation, performing slightly worse than the iERA5, despite the lower error variability ranges. The distributions of the RMSEs of iERA5 and

BMlinear, BM1, and BM10, show somewhat similar results for the median and overall variability. In this case, BMdense presents the best results. Finally, the ROCSS shows that all four DL architectures provide a better skill at representing Pr over the Iberian Peninsula, in comparison with iERA5, with median values ranging between 0.82 and 0.86, contrasting with 0.67 for the iERA5.

The results from Figs. 2 to 5 show that the four DL architectures are successful to downscale temperature and precipitation

from ERA5 at high resolution, presenting, in the vast majority of instances, a better performance than iERA5. Given the similar behavior of the four architectures, choosing the "best" one is not straightforward. BM10 and BMdense show the best precision (RMSEs) for the four variables (with BMdense being the most precise). However, considering the biases, BMdense produces the best results for $10^{th}$ percentile Tmin, BMlinear comes first for the $2^{nd}$ percentile of T, and BM10 produces more accurate results for Tmax. Regarding Pr, BMdense and BM1 retain the best performance for the mean and

98th percentile. Therefore, a clear distinction between architectures for all variables is not meaningful. Therefore, and assuming that all DL architectures are able to partially contribute to the overall performance of the downscaled datasets, the ensemble-building process considers equally all DL downscaling for each ESGCM.



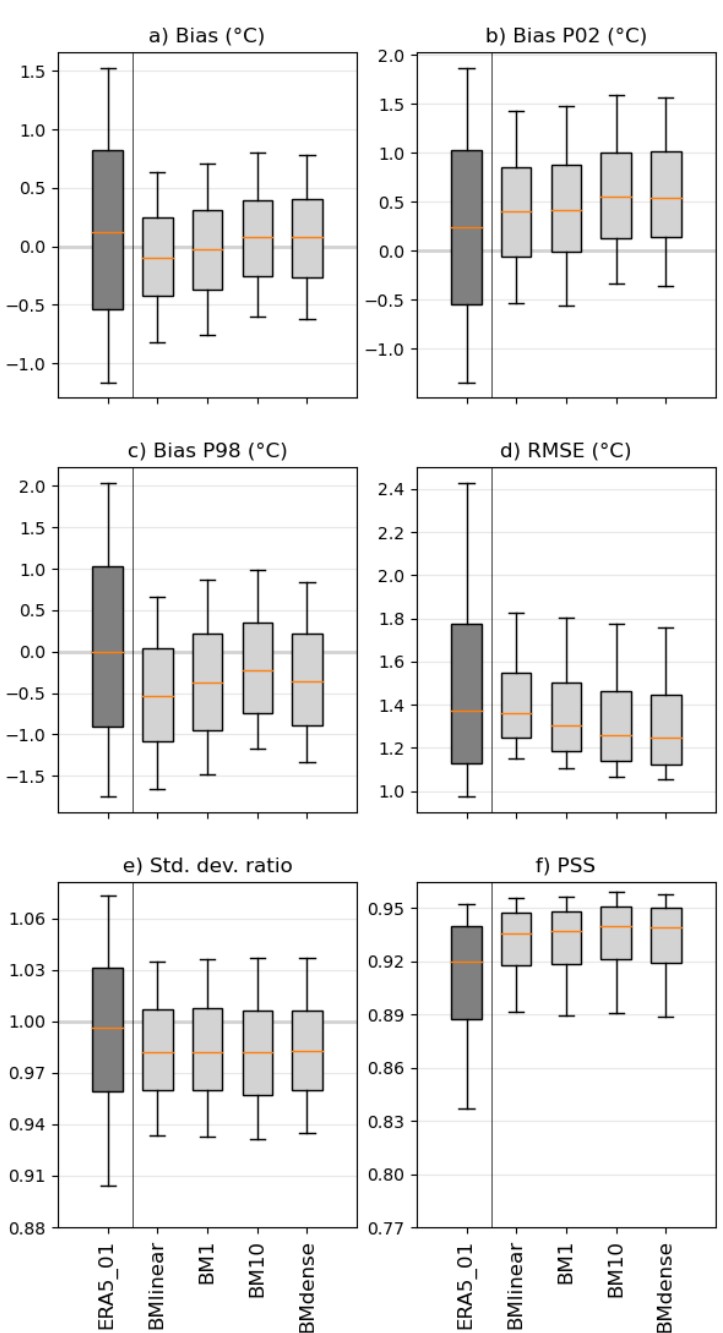

**Figure 2.** Error measures of the DL downscaling of ERA5 for the daily mean temperature (2010-2014) in relation to the Iberia01 observations. The errors considered are bias, bias of the 2nd and 98th percentiles, root mean square error (RMSE), standard deviation ratio and Perkins skill score (PSS). As reference the errors of ERA-5 interpolated to 0.1° are also shown.





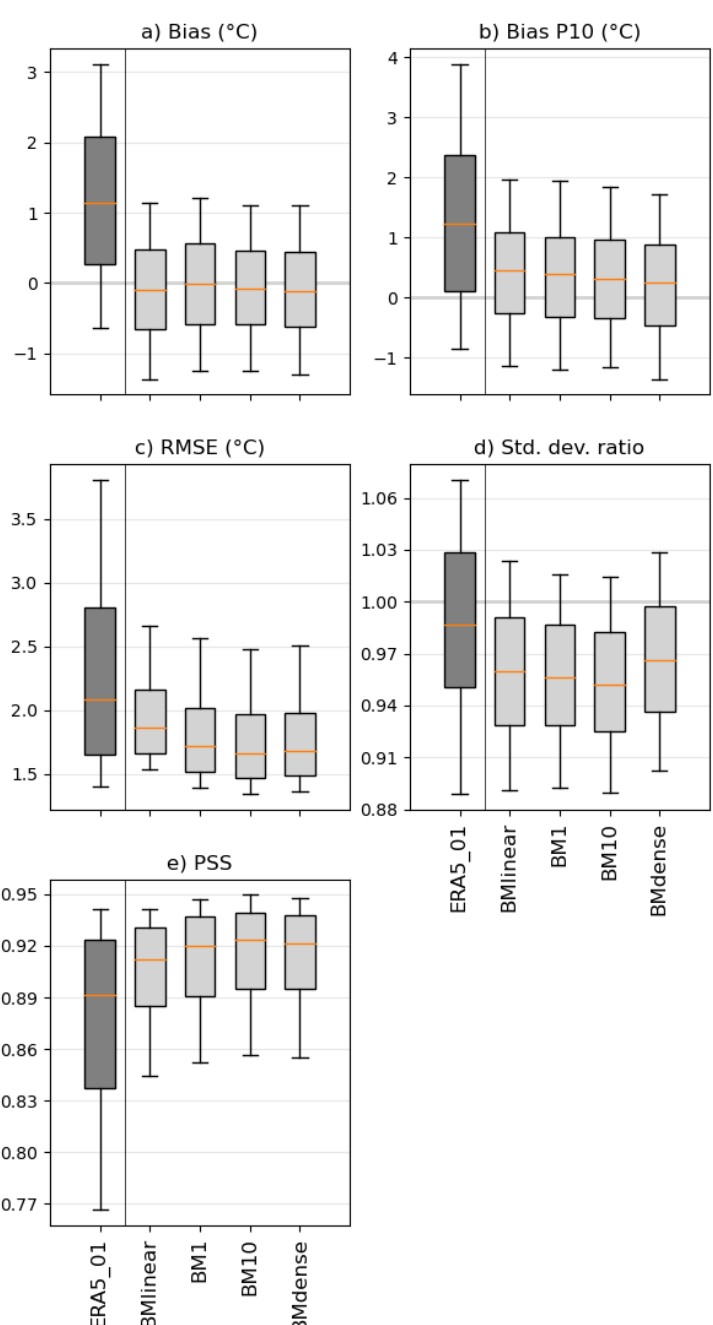

**Figure 3.** Error measures of the DL downscaling of ERA5 for the daily minimum temperature (2010-2014) in relation to the Iberia01 observations. The errors considered are bias, bias of the 10th percentile, root mean square error (RMSE), standard deviation ratio and Perkins skill score (PSS). As reference the errors of ERA-5 interpolated to 0.1° are also shown.



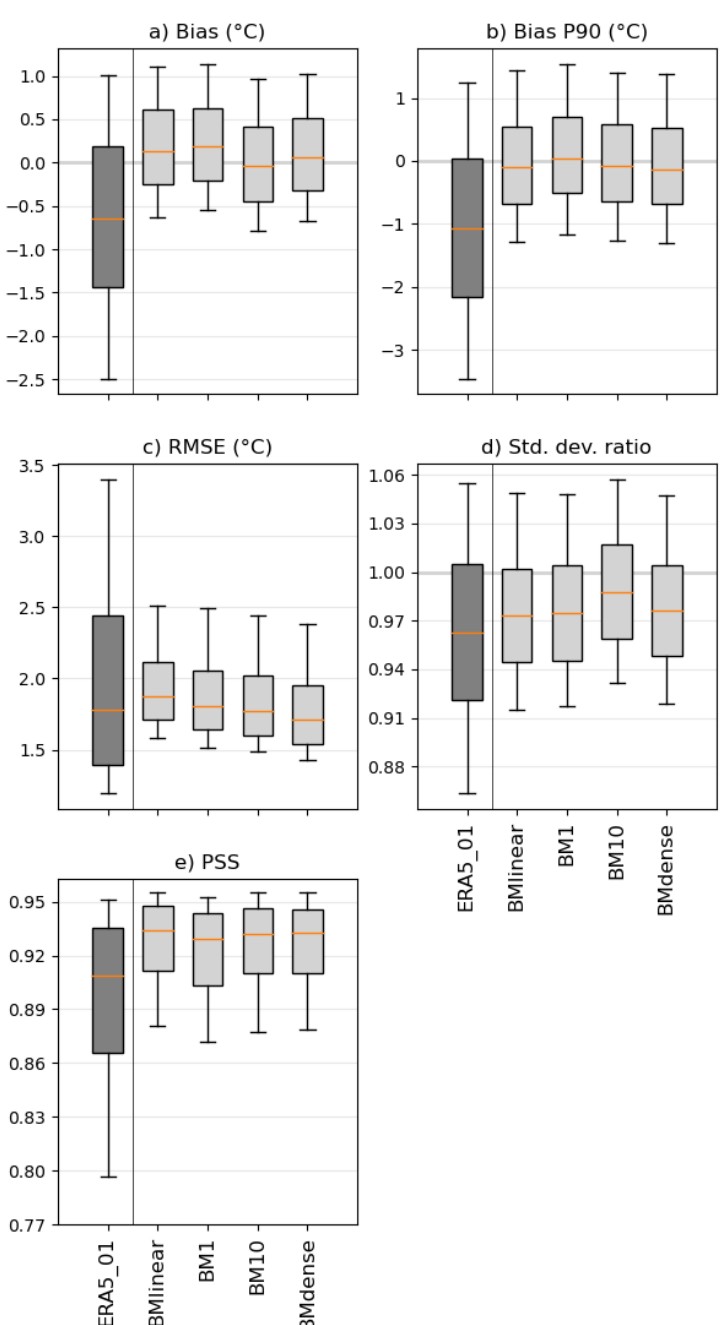

**Figure 4.** Error measures of the DL downscaling of ERA5 for daily maximum temperature (2010-2014) in relation to the Iberia01 observations. The errors considered are bias, bias of the 90th percentile, root mean square error (RMSE), standard deviation ratio and Perkins skill score (PSS). As reference the errors of ERA-5 interpolated to 0.1° are also shown.





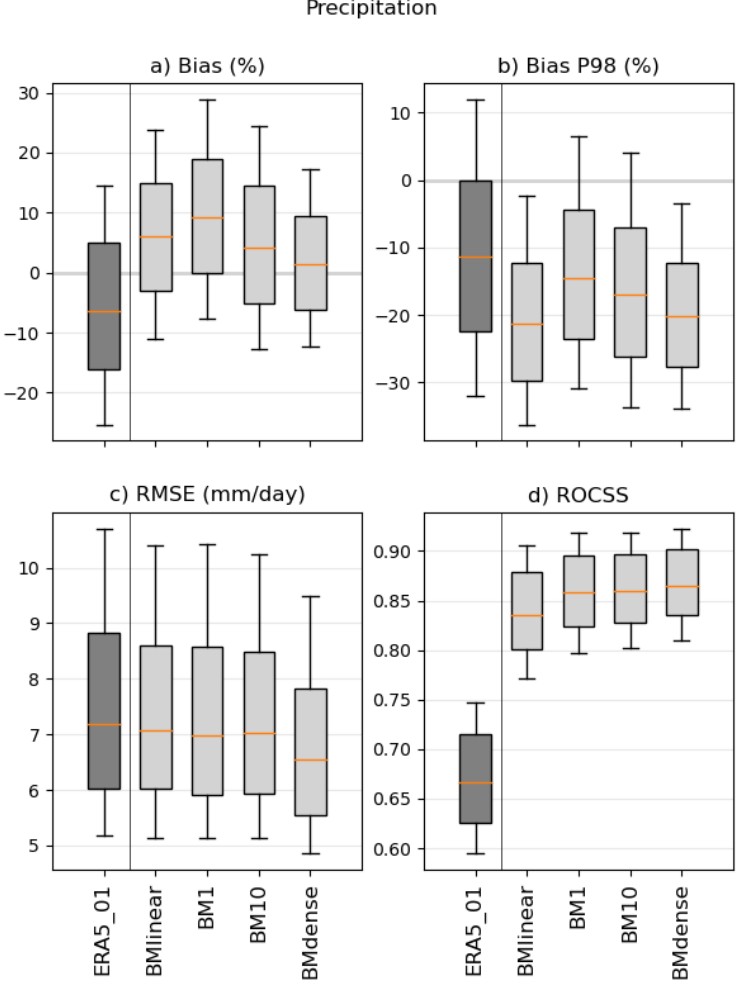

**Figure 5.** Error measures of the DL downscaling of ERA5 for daily precipitation (2010-2014) in relation to the Iberia01 observations. The errors considered are bias, bias of the 98th percentile, root mean square error (RMSE), and ROC skill score (ROCSS). As reference the errors of ERA-5 interpolated to 0.1° are also shown.

**3.2 Evaluation of DL forced by the ESGCMs**

In this section, the error metrics comparing the DL downscaling's of the CMIP6 ESGCMs and Iberia01 are displayed, for the four analysed variables (T, Tmin, Tmax and Pr), and evaluated in the context of the baseline dataset errors, like: the CMIP6 ESGCMs at 1°, and ERA5 (interpolated at both 1° and 0.1°, henceforth iERA5-1 and iERA5-0.1, or simply iERA5). Note that, for each DL downscaled ESGCM, a 4-member ensemble is considered, comprising the results from the four DL

architectures.

In general, for T (Fig. 6), the DL ESGCMs show a much better performance in comparison to the ESGCMs, and even w.r.t. the ERA5, at 0.1°. All biases for the DL ESGCMs results are around zero and show small variabilities (below 0.3ºC), The





forcing ESGCMs display both positive and negative median values for the three biases (total, 2nd and 98th percentiles'),
ranging in general between -1ºC and 2ºC, but some rise to 3ºC. The medians for iERA5 are generally closer to zero (below

0.3ºC). Regarding the RMSE, the DL downscaled ESGCMs show similar values, below 0.5ºC, while the iERA5 values are
typically below 1ºC, and most of the ESGCMs reach almost 4ºC, except for MIROC6, which exceeds this threshold. The
SDR of all models is around 1, nevertheless, the DL ensembles for each ESCGM present less variability. Finally, the PSS
metric shows that the DL ESGCMs are able to represent the Iberia01 PDFs remarkably well, yielding scores above 0.93. The
ESGCMs display median PSS values between 0.8 and 0.9 and are characterized by large variability.

Considering Tmin (Fig. 7), the biases for the DL ESGCMs results are around zero (ranging no more than 0.5ºC), while the
forcing ESGCMs and iERA5 show mainly positive values, with medians reaching 4.5°C and 2°C, respectively. The error
variability range for the DL ESGCMs is considerably smaller than for the ESGCMs counterparts. For the extreme Tmin
values (10th percentile), a similar pattern is visible, however with slightly greater biases for the ESGCMs. Regarding
RMSEs, the DL ESGCMs ensemble shows a great improvement with values around 1°C whereas the medians of ESGCMs

and ERA5 exceed 4°C and reach ~2°C, respectively, accompanied by much larger variability ranges. In terms of SDR, all
DL downscaling medians are near 1 and with rather small interquartile ranges when compared with ESGCMs and iERA5,
reaching 0.3 units. Finally, the PSSs metric consistently reveals the added value of the DL ensemble in representing the
PDFs with values ~0.94 that compare with values in the range of 0.70 and 0.87 of the ESGCMs.

Regarding Tmax (Fig. 8), the DL ensemble show a clear improvement w.r.t. to the forcing ESCGMs, with median biases less

than -0.2ºC, compared with a general underestimation of median values that reach 4°C for Tmax and its 90th percentile. The
MIROC6 is the only model overestimating Tmax in ~1ºC. The RMSE values display a striking reduction given by the DL
approaches, from median RMSE values ranging from 4ºC and 1.5ºC to less than 0.5ºC. The SDRs are closer to 1 than the
ESGCM counterparts as well as the iERA5. Considering the PSS, similarly to what was previously shown, the DL
downscaled inter-member variability ranges between 0.92 and 0.97, contrasting with the forcing ESGCMs and iERA5

(although the median PSSs for iERA5 are also high, above 0.9).

Finally, for precipitation (Fig. 9), the performance of the DL ensembles for each ESGCM is less remarkable then for
temperatures. Nevertheless, the DL downscalings outperform the forcing counterparts in all the error metrics, presenting
lower errors and variability ranges. Biases, both for Pr and its 98th percentile, point to a general underestimation, ranging
between -25% and 10%, yet corresponding to much lower overall differences in comparison to Iberia01 than the ESGCMs

and iERA5. For the RMSE, the DL ESGCMs and iERA5 are relatively equivalent, with median values of about 10 mm/day.
However, the ESGCMs RMSEs show values above 70 mm/day, but with medians between 18 e 35 mm/day. A similar
behaviour is identifiable for the ROCSS, with good results for both the DL ESCGMs and iERA5, with most median values
above 0.95, whether the ESGCMs median ROCSSs are in the range of 0.7 and 0.9, and extreme values reach -0.2. In
contrast, the PSSs of the DL downscaled ESGCMs show lower values, with medians around 0.5, smaller than the ~0.72 of

the ESGCMs. In some sense this is not that surprising since we are comparing the ESGCMs and Iberia01 precipitation at 1º,
which has a much smoother spatial pattern than at 0.1.



2-meter mean temperature

**Figure 6.** Error measures of the DL downscaling of CMIP6 ESGCMs for daily mean temperature (1979-2014) in relation to the Iberia01 observations. The errors considered are bias, bias of the 2nd and 98th percentile, root mean square error (RMSE), standard deviation ratio, and Perkins skill score. As reference the errors of ERA-5 interpolated to 1° and 0.1° and the errors of CMIP6 ESGCMs at 1° are also shown.






**2-meter minimum temperature**

**Figure 7.** Error measures of the DL downscaling of CMIP6 ESGCMs for daily minimum temperature (1979-2014) in relation to the
Iberia01 observations. The errors considered are bias, bias of the 10th percentile, root mean square error (RMSE) and Perkins skill score
(PSS), and standard deviation ratio. As reference the errors of ERA-5 interpolated to 1° and 0.1° and the errors of CMIP6 ESGCMs at 1°
are also shown.







**Figure 8.** Error measures of the DL downscaling of CMIP6 ESGCMs for daily maximum temperature (1979-2014) in relation to the Iberia01 observations. The errors considered are bias, bias of the 90th percentile, root mean square error (RMSE), standard deviation ratio, and Perkins skill score. As reference the errors of ERA-5 interpolated to 1° and 0.1° and the errors of CMIP6 ESGCMs at 1° are also shown.



**Figure 9.** Error measures of the DL downscaling of CMIP6 ESGCMs for daily precipitation (1979-2014) in relation to the Iberia01 observations. The errors considered are bias, bias of the 98th percentile, root mean square error (RMSE) and Perkins skill score, and ROC skill score. As reference the errors of ERA-5 interpolated to 1° and 0.1° and the errors of CMIP6 ESGCMs at 1° are also shown.





### 3.3 Iberian future mean climate

The evaluation of the DL architectures' ability to downscale both the ERA5 and the ESGCMs during the historical climate provided the necessary confidence to apply this method to downscale the future ESGCMs climate simulations. Therefore, here, the projected changes from the DL downscaled ESGCM ensemble are shown, obtained from the comparison of three

future time-slices (2015-2040, 2041-2070 and 2071-2100) with the 1981-2010 historical period, in terms of anomalies (*i.e.*, future *minus* historical). The four SSP-RCP pairs are analysed (SSP1-2.6, SSP2-4.5, SSP3-7.0, and SSP5-8.5), for each of the four variables. The democratic unweighted ensembles were built considering all ESCGMs and DL architectures. Therefore, the DL ESCGM ensembles are composed of 28 members (7 members times 4 architectures). Figures 10 to 13 refer to the projections for T, Tmin, Tmax and Pr, respectively. If less than two-thirds of the ESGCMs members agree on the

climate change signal, the grid-point is signalized with a grey dot, which reveals the lack of robustness of the projected change. A spatial comparison between the projected changes from the 1º ESGCMs ensemble, the 0.1º DL downscaled ESGCM ensemble, and the interpolated version of the latter, at 1º (to offer a fair comparison with the original datasets), is conducted, to highlight the differences and added value brought by the DL downscaled ensembles.

The future projected changes for T are displayed in Fig. 10, for the forcing 1º ESGCMs ("1º GCM" in the panels) and for the

DL downscaled ESGCM ensemble ("DL-MM_01" in the panels), for the three future time-slices under the four scenarios. Overall, the results show a projected increase in T, starting from the 2015-2040 period and continuing towards the end of the 21st century (Fig. 10a). Naturally, the SSP1-2.6 (SSP5-8.5) scenario depicts the smallest (greatest) changes. Under the SSP1-2.6, projected changes of up to 2.5ºC are discernible, and the patterns exhibit analogous characteristics when comparing the ensemble of ESGCM to the downscaled ensembles generated using DL (Fig. 10a). This similarity is also evident in the

remaining scenarios, albeit with the additional advantage of DL downscaled ESGCM ensembles displaying more detailed patterns of warming. Both DL and ESGCM ensembles demonstrate temperature increases of up to 1.5ºC, 3.5ºC, and 6ºC during the periods 2015-2040, 2041-2070, and 2071-2100, respectively, under the SSP5-8.5 scenario. But the corresponding median warming values for Iberia are around 1.23ºC, 2.5ºC and 5ºC. In the case of SSP2-4.5 and SSP3-7.0, there is less pronounced warming, although it may still reach up 3.5. and 4.5ºC, respectively. These results are more easily observed by

condensing the spatial information into boxplots (Fig. 10b). Overall, differences between DL and ESGCM ensemble are more pronounced from the middle of the century onwards, especially for the two worst-case scenarios (SSP3-7.0 and SSP5-8.5).

Considering Tmin (Fig. 11), results present similar features to those from T. Within the SSP1-2.6, projected changes between 0.5ºC and 2ºC are visible, with more pronounced warming in the end of the century. The behavior is similar

between the ESGCM ensemble and the downscaled one. Local variations in the patterns of Tmin projected changes are visible for all time-slices and scenarios in the outcomes from the DL downscaled ensemble, compatible with the results from higher-resolution models, able to describe local phenomena in greater detail (contrarily to a simple interpolation method). For the SSP5-8.5 scenario, results for the 2041–2070 (2071–2100) period are similar between ensembles, with projected





increases from 2ºC to 3.5ºC (3ºC to 5.5ºC). Note, however, that the DL ensemble projects local increases of up to 6ºC in
central Iberian Peninsula, which are not present in the ESGCM ensemble projections. This behaviour is also depicted in the
boxplots of Fig. 11b.

Tmax (Fig. 12) presents similar characteristics to T and Tmin. In the beginning of the 21st century (2015-2040), the
magnitude of the projections from both the ESGCM and DL ensemble ranges from 0.5ºC to 2ºC in most of the Iberian
Peninsula (Fig. 12a), independently of the scenario. In the mid-21st century (2041-2070), projections from both the ESGCM
and DL downscaled ensembles represent a similar range of projected changes (up to 3.5ºC, depending on the scenario; Fig.
12b). By 2071-2100, warming values are almost twofold than those of the middle of the century, surpassing 6.5ºC in the
worst-case scenario. It should be highlighted that the DL downscaled ensemble shows different areas of extreme projected
increases in Tmax (towards south), that are not present in the ESGCM ensemble (where the largest warming is found
towards more central and northeastern regions).

Figure 13 shows the Pr projected changes for the future time-slices and scenarios, in this case, considering the mean daily
accumulated values, and their changes, in mm/day. In opposite manner, the changes depicted in Fig. 13 are rather different
from the ones in Figs. 10 to 12. While the ESGCM ensemble projects rather homogeneous decrease in the mean daily
precipitation for all future periods and scenarios, the DL downscaled ensemble show mostly consistent decreases in the
western and north areas of Iberia, and non-robust regional increases throughout central and eastern Iberia, independently of
the period and scenario. It is important to emphasize that most of these projected increases are not robust (*i.e.*, less than ⅔ of
the ensemble members agree on the signal), whereas almost all projected decreases are (Fig. 13a). Negative Pr projections
are found mainly in the northern, western and southwestern portions of Iberia, increasing in area and robustness towards
2100, and with the SSP5-8.5 scenario. These features are in overall agreement with the ESGCM ensemble, nevertheless,
with much increased detail due to the enhanced horizontal resolution. In fact, for the 2071-2100 time-slice under the SSP5-
8.5, the ESCGM (DL downscaled) ensemble shows projected decreases of down to -0.75 mm/day (-1 mm/day in the
northern and northwestern Iberia). The boxplots in Fig. 13b are largely affected by the compensating effect of different
signal projected changes, resulting in overall larger ranges of projected change (even for the interpolated DL ensemble, at
1º), and median values closer to zero, in comparison with the ESGCM ensemble. Nonetheless, an overall decrease of Iberian
precipitation is visible, that for the DL ensemble is smaller than the one shown by the forcing ESCGM ensemble.

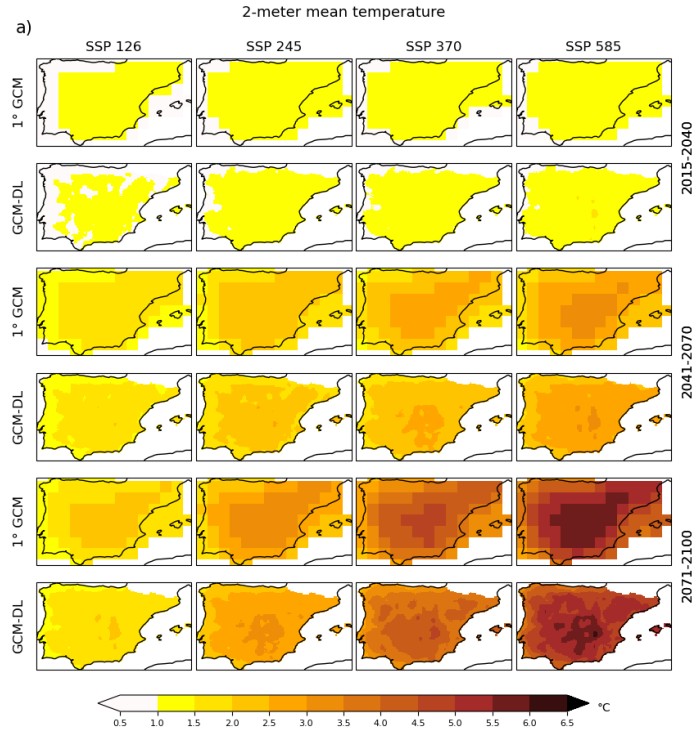

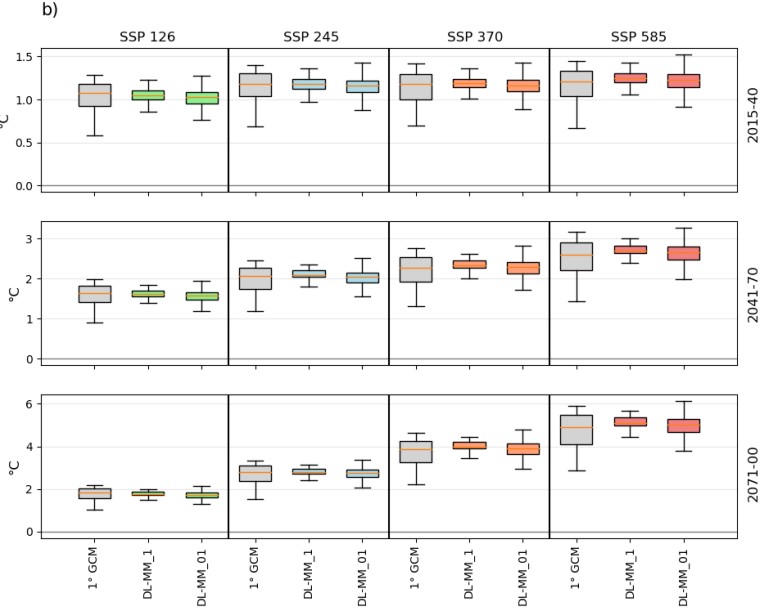

**Figure 10.** Mean temperature relative changes given by the DL CMIP6 ESGCMs multi-model ensemble at 0.1° for SSP1-2.6, SSP2-4.5, SSP3-7.0 and SSP5-8.5, (2015-2040, 2041-2070, 2071–2100 minus 1981–2010)/1981–2100. a) Maps. Grey dots specify gridpoints where less than two-thirds of the DL-CMIP6 ESGCMs pairs agree on the climate change signal (no occurrences). b) Boxplots. The DL CMIP6 ESGCMs multi-model ensemble were interpolated to 1° and the results are also displayed. As reference, the climate change signal linked to the ESGCMs ensemble at 1° is also shown in a) and b).



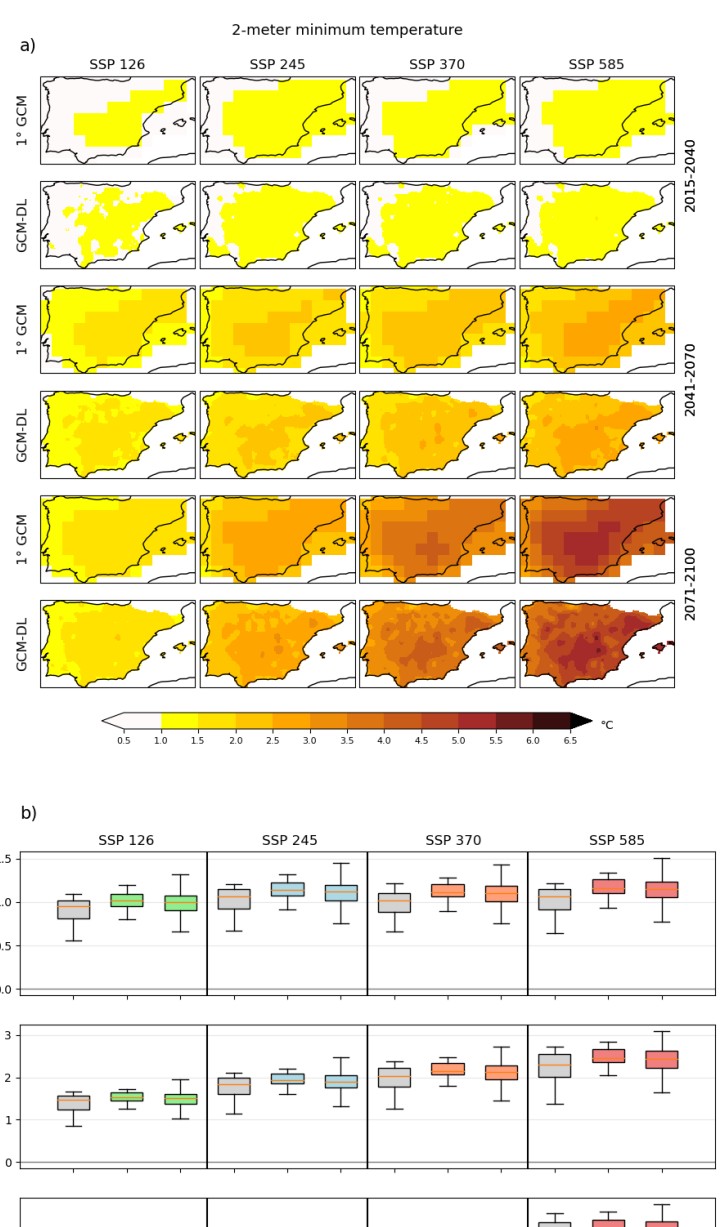

**Figure 11.** Minimum temperature relative changes given by the DL CMIP6 ESGCMs multi-model ensemble at 0.1° for SSP1-2.6, SSP2-4.5, SSP3-7.0 and SSP5-8.5, (2015-2040, 2041-2070, 2071–2100 minus 1981–2010)/1981–2100. a) Maps. Grey dots specify gridpoints where less than two-thirds of the DL-CMIP6 ESGCMs pairs agree on the climate change signal (no occurrences). b) Boxplots. The DL CMIP6 ESGCMs multi-model ensemble were interpolated to 1° and the results are also displayed. As reference, the climate change signal linked to the ESGCMs ensemble at 1° is also shown in a) and b).





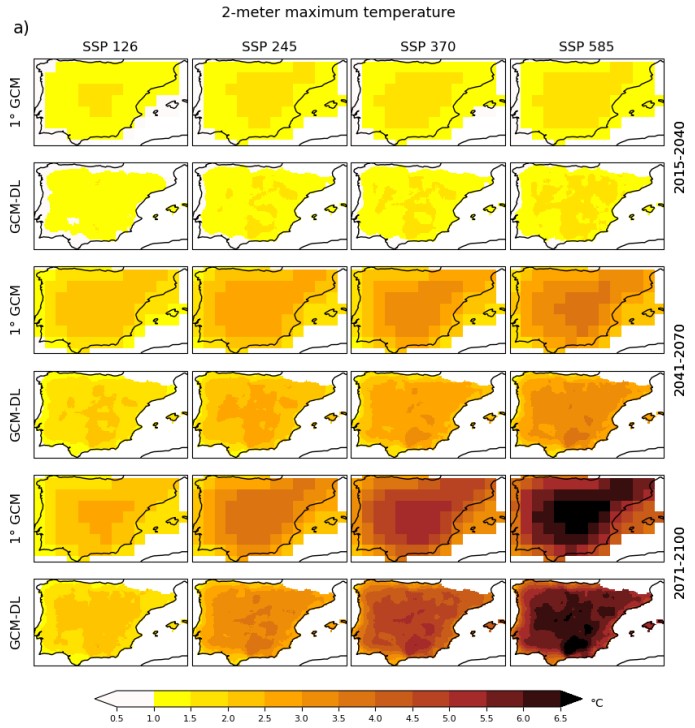

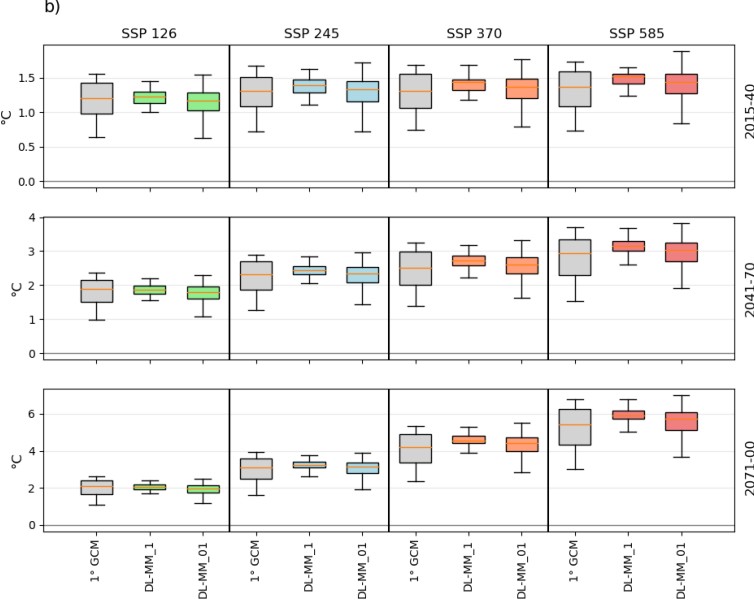

**Figure 12.** Maximum temperature relative changes given by the DL CMIP6 ESGCMs multi-model ensemble at 0.1° for SSP1-2.6, SSP2-4.5, SSP3-7.0 and SSP5-8.5, (2015-2040, 2041-2070, 2071–2100 minus 1981–2010)/1981–2100. a) Maps. Grey dots specify gridpoints where less than two-thirds of the DL-CMIP6 ESGCMs pairs agree on the climate change signal (no occurrences). b) Boxplots. The DL CMIP6 ESGCMs multi-model ensemble were interpolated to 1° and the results are also displayed. As reference, the climate change signal linked to the ESGCMs ensemble at 1° is also shown in a) and b).



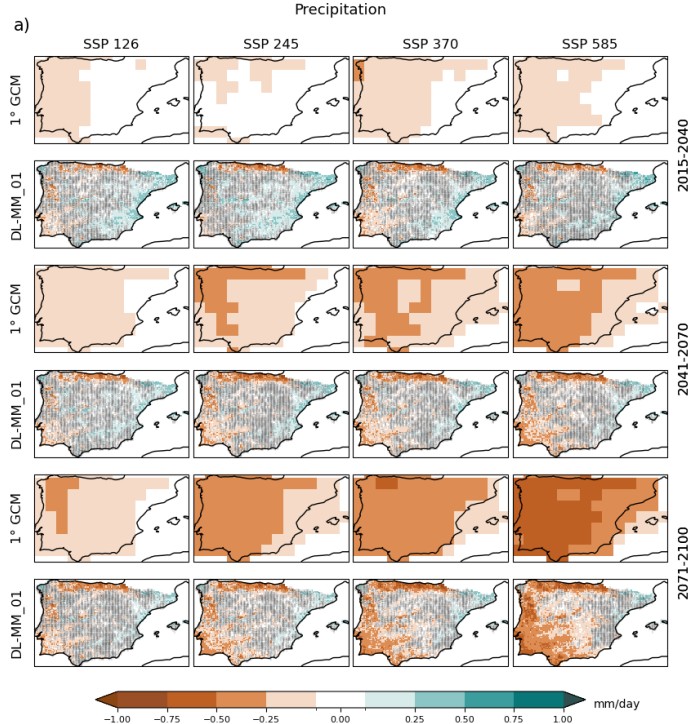

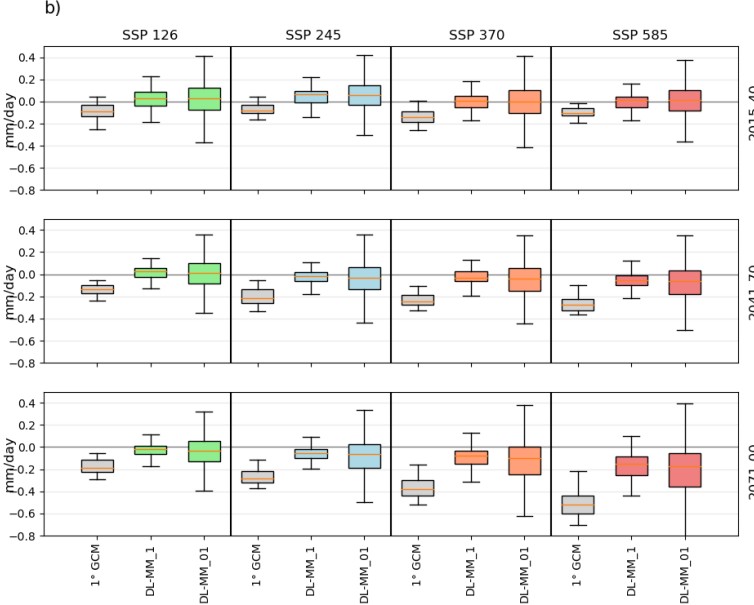

**Figure 13.** Daily mean precipitation relative changes given by the DL CMIP6 ESGCMs multi-model ensemble at 0.1° for SSP1-2.6, SSP2-4.5, SSP3-7.0 and SSP5-8.5, (2015-2040, 2041-2070, 2071–2100 minus 1981–2010)/1981–2100. a) Maps. Grey dots specify gridpoints where less than two-thirds of the DL-CMIP6 ESGCMs pairs agree on the climate change signal (no occurrences). b) Boxplots. The DL CMIP6 ESGCMs multi-model ensemble were interpolated to 1° and the results are also displayed. As reference, the climate change signal linked to the ESGCMs ensemble at 1° is also shown in a) and b).





### 3.4 Iberian future climate extremes

Considering climate extremes, in this section, the projected changes of three climate extreme indices are compared for both the ESGCM and the DL downscaled ESCGM ensembles, similar to section 3.3. For the Tmin and Tmax, the 10th and 90th percentiles were considered, respectively, while, for the extreme precipitation, the 95th percentile of the daily mean accumulated values was computed.

In general, the future 10th percentile of Tmin (Fig. 14) reveals lower warming projections than for Tmin (Fig. 11) and also
different patterns. The most pronounced warmings are located in the south-central and eastern regions of Iberia (Fig.14a), which may reach 2°C (4°C) in the 2041-2070 (2071-2100) period, for the SSP5-8.5 scenario. The remaining scenarios show lower warmings, reaching 1.5°C, 2.5°C, and 3.5°C for SSP1-2.6, SSP2-4.5, and SSP3-7.0, respectively, by the end of the century. As expected, the warming patterns are much more detailed and localized when the DL ensemble is considered.

Similar to extreme Tmin, the projections of extreme Tmax (Fig. 15) exhibit comparable patterns to those of the mean climate
variable (Fig. 12), albeit with much more pronounced warming values. In particular, over a significant portion of Iberia, the warming reaches over 8°C by the end of the century for SSP5-8.5 scenario. The use of a precise and performance-evaluated technique to downscale a large ensemble of ESGCM climate projections at a high resolution provides substantial added value in capturing local climate change for the 90th percentile of Tmax, as demonstrated in Fig. 15a. For instance, when considering the 2071-2100 period under SSP5-8.5, both the ESGCM and DL downscaled ensembles project changes
exceeding 8°C. However, the DL downscaled ensemble surpasses this threshold over a wider area, locally exceeding 9°C and even extending to the southern coast of Iberia, where the projections from the ESGCM ensemble do not surpass 6°C.

Regarding the extreme Pr (Fig. 16), the DL projections point to reductions of the extreme precipitation across southwestern Iberia, expanding eastward (for part of southern Iberia) throughout the 21st century, and more pronounced for SSP3-7.0 and SSP5-8.5 scenarios. These decreases can reach more than 20 mm/day in these regions. On the other hand, essentially over
central, southeastern and northwestern Iberia, DL projections show an intensification in extreme precipitation in all scenarios and time periods, reaching increases that surpass 20 mm/day. The ESGCMs projections mostly present decreases in extreme precipitation (for all time-slices and scenarios except the SSP1-2.6 during 2015-2040 and 2041-2070 in the northern half of the Peninsula; Fig. 16a). The spatial pattern of changes in extreme precipitation are dissimilar for the DL and ESGCMs projections. The boxplots in Fig. 16b summarize the differences between the ESGCM and DL downscaled ensembles, being,
nevertheless, slightly affected by the lack of robustness of some of the outcomes – the DL ensemble shows a much larger variability than the ESGCM ensemble.



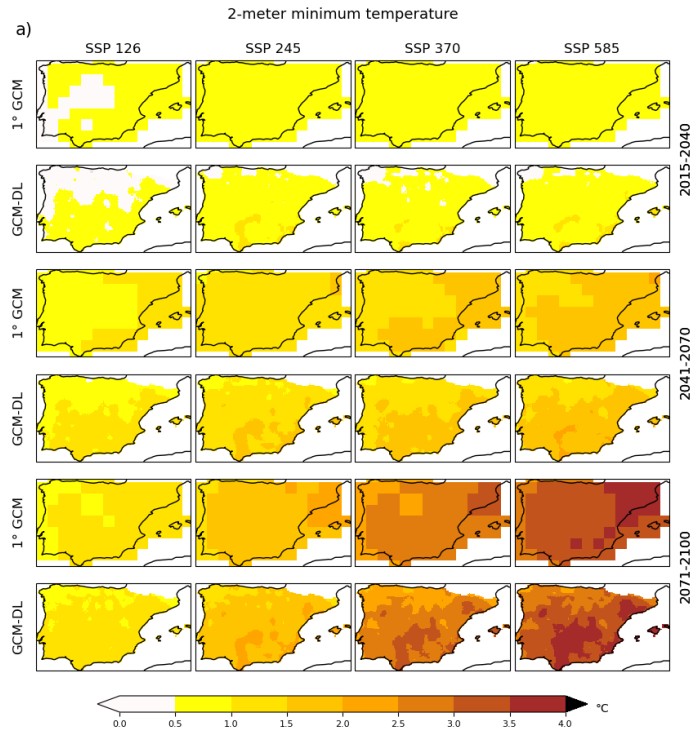

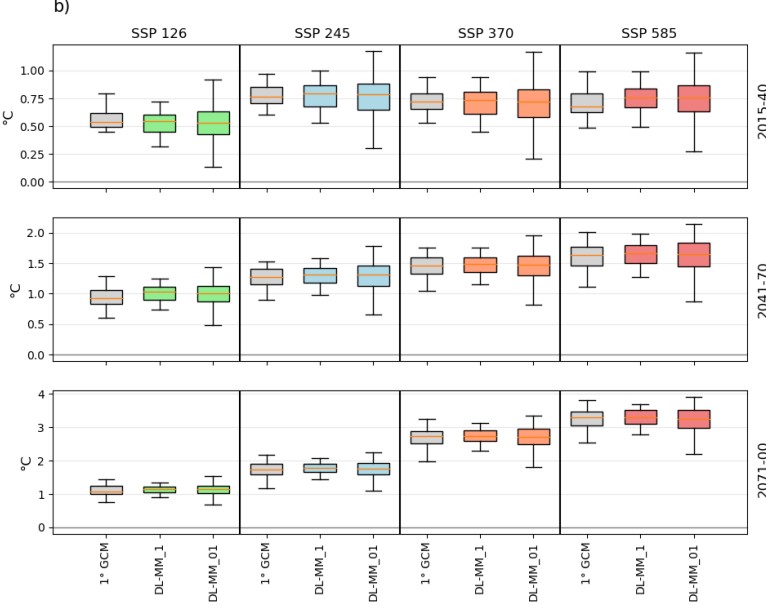

**Figure 14.** Mean minimum temperature 10th percentile relative changes given by the DL CMIP6 ESGCMs multi-model ensemble at 0.1° for SSP1-2.6, SSP2-4.5, SSP3-7.0 and SSP5-8.5, (2015-2040, 2041-2070, 2071–2100 minus 1981–2010)/1981–2010. Grey dots represent gridpoints where less than two-thirds of the DL-CMIP6 ESGCMs pairs agree on the climate change signal (no occurrences). As reference the climate change signal linked to all ESGCMs at 1° is also shown.



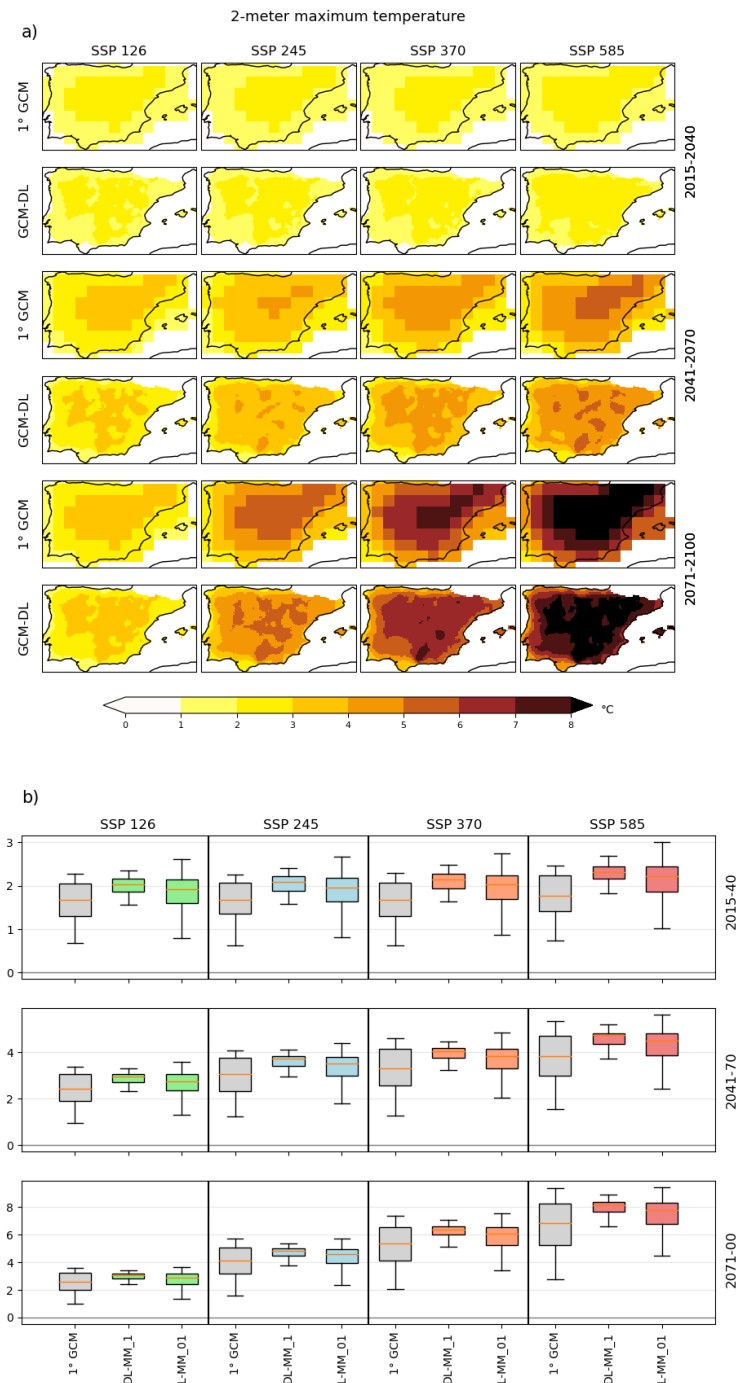

**Figure 15.** Mean maximum temperature 90th percentile relative changes given by the DL CMIP6 ESGCMs multi-model ensemble at 0.1°
for SSP1-2.6, SSP2-4.5, SSP3-7.0 and SSP5-8.5, (2015-2040, 2041-2070, 2071–2100 minus 1981–2010)/1981–2010. Grey dots represent
gridpoints where less than two-thirds of the DL-CMIP6 ESGCMs pairs agree on the climate change signal (no occurrences). As reference
the climate change signal linked to all ESGCMs at 1° is also shown.

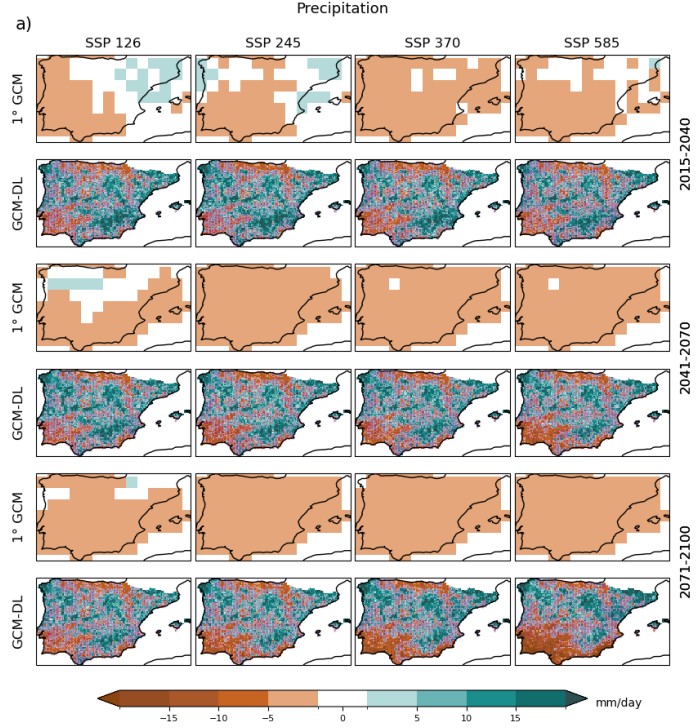

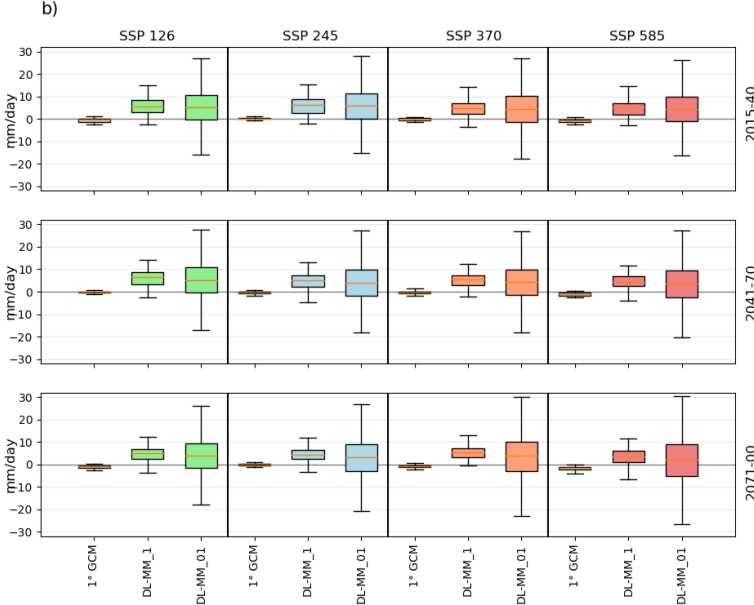

**Figure 16.** Precipitation 95th percentile relative changes given by the DL CMIP6 ESGCMs, multi-model ensemble at 0.1° for SSP1-2.6, SSP2-4.5, SSP3-7.0 and SSP5-8.5, (2015-2040, 2041-2070, 2071–2100 minus 1981–2010)/1981–2010. Purple dots represent gridpoints where less than two-thirds of the DL-CMIP6 ESGCMs pairs agree on the climate change signal. As reference the climate change signal linked to all ESGCMs is also shown.





## 4. Discussion and Conclusions

The Iberian Peninsula, situated in the southwestern tip of the European continent, within the Mediterranean region, is considered a climate change hotspot, due to the projected future warming and drying conditions. These changes can
significantly impact the natural environment and human health in the region (Giorgi, 2006; Soares et al., 2017; Cramer et al., 2018; Lionello and Scarascia, 2018; Cardoso et al., 2019; Tuel and Eltahir, 2020; Soares and Lima, 2022; Lima et al., 2023a,b; Soares et al., 2023). Consequently, there is an urgent need for accurate climate information to assist the planning and development of adaptation strategies. Recent climate change studies focusing on the Iberian Peninsula relied on RCM simulations forced by CMIP5 GCMs (Soares et al., 2017; Cardoso et al., 2019; Lima et al., 2023a,b; Soares et al., 2023) to
project future climate change with increased resolution, accounting for regional features not captured by coarse ESGCMs. However, following the release of the improved CMIP6 global climate simulations and projections (in the context of the most recent IPCC report: AR6; IPCC, 2019), the need for an updated climate change assessment in the Iberia Peninsula arose. The new high-resolution CMIP6 EURO-CORDEX regional climate simulations and projections will become available within one to two years. In the interim, however, there is a need for high-resolution climate information to accurately assess
future projections over Iberia. In this context, an opportunity emerges, to explore alternative approaches to downscale the current CMIP6 simulations and projections. Therefore, this study leverages innovative AI methods to evaluate the evolution of mean, minimum, and maximum temperatures, as well as precipitation, across the Iberian Peninsula, throughout the 21$^{st}$ century. The analysis is based on a multi-model, multi-architecture ensemble of CNN-based downscaled projections derived from CMIP6 ESGCMs. The investigation encompasses three distinct future time-slices (2015-2040, 2041-2070, 2071-2100)
in line with four SSPs-RCPs scenarios: SSP1-2.6, SSP2-4.5, SSP3-7.0, and SSP5-8.5.

On a first instance, the ability of four DL architectures to reproduce the historical T, Tmin, Tmax and Pr climates was evaluated over Iberia during 2010-2014 (Figs. 2 to 5). During this period, all DL architectures, trained using ERA5 data between 1979 and 2004, revealed a good agreement with observations (Iberia01) for the predictand variables (using solely the predictors as input data). Although more complex architectures, such as the BMdense, revealed better performance for Pr
(lower overall biases and RMSEs, and higher ROCSS), a clear distinction between architectures was not meaningful. Therefore, we opted to consider all four DL architectures to downscale the CMIP6 ESGCMs, obtaining a 4-member ensemble per model. The results showed that during the 1979-2014 historical period, the DL downscaled ESGCMs were able to represent the Iberia01 reference climate with large increased performance in comparison with the forcing ESGCMs, and even compared with the ERA5 and iERA5 datasets (Figs. 6 to 9). For Pr, nevertheless, the downscaled error metrics
were shown to be similar to the reanalysis' ones, despite greater differences in the overall variable distributions (as shown by the PSS values). Such disagreement could be related to the singular behaviour of Pr, especially considering its extreme events, which can occur under distinct atmospheric synoptic patterns (predictor sets), becoming challenging for the DL architectures to establish empirical relationships between the vertical atmospheric structure and surface level precipitation accumulation. Overall, the evaluation of the DL downscaled ESGCMs showed a rather good performance in representing the



historical climate (mean, minimum and maximum temperatures, and precipitation), providing the necessary confidence to project the future climate change under different scenarios using this new approach.

The DL downscaled T projections revealed a projected increase between 1ºC and 1.5ºC over Iberia (Fig. 10), for all scenarios, during the earliest future period (2011-2040). By the end of the 21$^{st}$ century, nevertheless, the DL ensemble projected changes were shown to become more heterogeneous between scenarios, generally varying between 1.5ºC (SSP1-

2.6) and 5ºC (SSP5-8.5). In all instances, the DL downscaled T projected changes showed a strong agreement with the original CMIP6 ESGCM ensemble, in both the signal and main spatial patterns of climate change. Nevertheless, regional-to-local features are clearly enhanced by the increased resolution. In fact, the most poignant difference between the DL and the original ensembles is the horizontal discretization. Local differences can be identified in the most inland areas of the Iberian Peninsula in the DL results, which are not captured by the original ESGCMs, due to the coarse grid, neglecting valleys and

other geographically enclosed areas, which foster greater horizontal heterogeneities. Similar features were found for Tmin (Fig. 11) and Tmax (Fig. 12), and for the respective extreme values (10$^{th}$ percentile of Tmin and 90$^{th}$ percentile of Tmax in Figs. 14 and 15, respectively). Between those, Tmax showed larger projected increases than Tmin (approximately twofold), revealing greater intra-daily temperature ranges to be expected in the future. For the extreme Tmax values, even under the SSP1-2.6 "optimistic" scenario, DL downscaled projections revealed increases exceeding 3ºC by the end of the 21$^{st}$ century.

For the "less optimistic" ones (SSP3-7.0 and SSP5-8.5), extreme Tmax projected changes of up to 7ºC were shown for most of Iberia (Fig 15). DL downscaled projections for extreme Tmin, on the other hand, were shown to be higher in the eastern and southern Iberia (Fig. 14), locally surpassing 3.5ºC (for 2071-2100 under SSP5-8.5). Overall, these projections are aligned with the EURO-CORDEX ensemble projections for Iberia (Soares et al., 2017; Cardoso et al., 2019, Lima et al., 2023a; Amblar et al., 2017), but with small value differences which are also linked to the dissimilarities regarding the

emission scenarios.

The significance of DL downscaling techniques in the context of Pr projections unveiled further intricacies when compared to T, Tmin, and Tmax projections. Given that the behaviour of daily mean accumulated precipitation is heavily influenced by local topography and other phenomena, particularly owing to convective processes, which can result in local, large precipitation accumulations, projecting Pr was shown to be more complex for DL downscaling methods, considering the

widespread continental, mountainous areas of the Iberian Peninsula. Therefore, for both Pr (Fig. 13) and extreme Pr (95th percentile; Fig. 16), the original and DL downscaled ESGCM ensemble projections showed greater discrepancies. While the ESGCM projected changes showed essentially negative values, corresponding to the future large-scale expected drying over Iberia, DL results revealed: a drying trend in western and south-western regions of Iberia, stronger for the upper-end scenarios, and, local projected increases, mostly in the central and eastern continental regions. The southern and western

precipitation reductions are consistent with a significant reduction of large-scale precipitation from frontal activity, due to the northern displacement of the storm-tracks (Tamarin et al., 2017). In fact, the northward expansion of the Hadley Cell lead to a northward shift of the storm tracks over the North Atlantic, resulting in the reduction of large-scale precipitation across southern and western of Iberia (Bengtsson et al., 2006; Harvey et al., 2014; Kang and Lu, 2012; Ulbrich et al.,



2008).The projected local increases of precipitation, although non-robust, mostly with less than ⅔ of agreement between ensemble members, may be consistent with local-to-regional changes in convective precipitation that are not captured by the original ESGCMs, highlighting potential applications of DL techniques to long-term projections (or even short-term forecasting).

The projected changes of warming and drying over Iberia, as reported in recent studies using previous CMIP outputs (CMIP5), as well as in the most recent IPCC report (IPCC, 2021), are consistent with the multi-model, multi-scenario, multi-architecture DL downscaled ESGCM ensemble projections presented in this study. This behaviour is also in accordance with the resulting DL climate change signals from Baño-Medina et al. (2022) for Iberia, which showed similar spatial patterns to those obtained from the CMIP5 RCMs, nevertheless, with local-to-regional added value. Previous research has demonstrated that the warming and drying trends over Iberia are more pronounced under high anthropogenic emission scenarios, reflecting the influence of human activities on climate change, compared to the natural variability of the climate system. Our results demonstrated that in the CMIP6 context, within the new set of scenarios encompassing socioeconomic and radiative concentration pathways, AI-based DL methods are able to accurately simulate the historical Iberian climate, and produce consistent high-resolution scenario-based projections, by the use of (coarse) GCM forcing and a high-resolution training database. Thus, the present study highlighted the substantial advantages of employing novel approaches based on DL to obtain efficiently up-to-date, high-resolution climate information at a local scale, specifically for Iberia. This is crucial for supporting and designing mitigation and adaptation strategies.

**Code and data availability**

The datasets generated and/or analysed during the current study are available through the Earth System Grid Federation (ESGF) Lawrence Livermore National Laboratory (LNLL) repository for CMIP6 ESGCMs (https://esgf-node.llnl.gov/projects/esgf-llnl/, last access: June 2023); and the Copernicus Climate Change Service (C3S) and Copernicus Data Store (CDS) repository for ERA5 (https://cds.climate.copernicus.eu/cdsapp#!/dataset/reanalysis-era5-pressure-levels?tab=overview and https://cds.climate.copernicus.eu/cdsapp#!/dataset/reanalysis-era5-single-levels?tab=overview, last access: June 2023; Copernicus Climate Change Service (C3S), 2019). The Iberia01 dataset is publicly available through the DIGITAL.CSIC open science service (Herrera et al., 2019, https://doi.org/10.20350/digitalCSIC/8641, last access: June 2023). The DL configuration is available upon request.

**Author contributions**

The conceptualization was done by PMMS. CMIP6 ESGCMs data were acquired and processed by DCAL and GL. ERA5 and Iberia01 data were acquired by FJ. The formal analysis and visualization were done by FJ. All authors contributed to the writing, reviewing, and editing of the paper.



**Competing interests**

The authors declare that they have no conflict of interest.

**Acknowledgements**

The authors would like to acknowledge the financial support of Portuguese Fundação para a Ciência e a Tecnologia (FCT) I.P./MCTES through national funds (PIDDAC) – UIDB/50019/2020 – Instituto Dom Luiz. The authors also acknowledge the EEA-Financial Mechanism 2014-2021 and the Portuguese Environment Agency through Pre-defined Project-2 National
Roadmap for Adaptation XXI (PDP-2) and project DHEFEUS – 2022.09185.PTDC.

**Financial support**

The publication of this research is funded by the Portuguese Fundação para a Ciência e a Tecnologia (FCT) I.P./MCTES through national funds (PIDDAC) – UIDB/50019/2020 – Instituto Dom Luiz.

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
