# Peer review of "High resolution downscaling of CMIP6 Earth System and Global Climate Models using deep learning for Iberia"

_Geoscientific Model Development, 2023_

## Author Comment (AC3)

**In this manuscript, deep learning methodology based on CNN architectures is evaluated for downscaling CMIP6 simulations and projection with standard resolutions to a higher resolution, i.e. 1/10th deg. The analysis focuses on the Iberian Peninsula, and the ability of four CNN architectures to reproduce the surface T, Tmin, Tmax and Pr climates is evaluated. The DL algorithm is trained using ERA5 and compared to the high-resolution regular gridded dataset over the historical period, and then used to downscale future projections (relative to historical climate) in agreement with four future scenarios and multi-models.**

**Overall, the manuscript is well organized in format and the writing is clear. Although the physical process is rarely touched (due to the limitation of DL), the downscaled results based on DL are sound. I have some comments regarding clarifications in the main text.**

We would like to thank the reviewer for their availability to read and comment our manuscript. We are grateful for your positive and constructive comments and suggestions. Our response to each comment is given below. We sincerely think that your revision allowed an overall improvement of the manuscript.

**Line 23: "Notably, …. climate scenario". Please specify the region (Iberia) exhibiting the temperature increase.**

We have now specified the region in the sentence: "*Notably, a clear warming trend is observed in Iberia*, … climate scenario*".*

**Line 129: I am not familiar with any paper using "democratic" to describe a simple average. It would be better to just use "simple average", which is straightforward and easy to follow. I noticed that "democratic" has been mentioned several times in this manuscript. Please also revise those instances accordingly.**

We have implemented the reviewer's recommendation by incorporating the suggested changes throughout the entire text, using the terminology "simple average".

**Line 145 (Fig. 1) Can you mention why this domain of predictors is chosen (dashed line)? Have you tested the sensitivity of the downscaled fields when the domain of predicators is changed?**

The chosen domain for the predictors is larger than the output domain to ensure that the necessary information about large-scale phenomena given by the predictors is provided to perform a solid downscaling. We made an initial test with a smaller domain, Iberia-only, similar to the output domain. The downscaling results were overall slightly inferior in quality when using a smaller domain. To maintain computational efficiency, we did not try downscaling with larger domains.

We added the following sentence in line 148: *"The predictors domain (Fig. 1, red dashed line) is a larger region than the IP domain to ensure that large-scale phenomena are included in the information provided by the predictors to train the DL models."*

**Lines 170-173: These sentences would be better placed in the introduction.**

The sentences were moved to the introduction (lines 113-116). We also added an introductory sentence about Iberia01 prior to those sentences (lines 111-113): "*The Iberia01 regular gridded product (hereafter Iberia01) is the highest resolution observational daily dataset including mean, maximum and minimum temperatures and precipitation, covering the full domain of continental Iberia (Herrera et al., 2019)."*

**Table 1: It would be better to provide the full name of CMIP6 ESGCM. For example, CM6A-LR -> IPSL-CM6A-LR.**

Full names provided.

**Lines 220-222: I am not able to follow this sentence. Could you rephrase it?**

We rephrased the sentence as follows: *"For precipitation, however, the DL models feature a multi-output structure (see Fig. 3 of Baño-Medina et al., 2020)."*

**Lines 222-223: How are these terms (alphas, beta, and distribution) used in this study? I couldn't find more discussions about these terms from CMIP6 runs.**

These terms are the output of the DL models used to downscale precipitation. The precipitation value is then obtained by multiplying the alpha and beta terms. The results shown are discussed based on the resulting precipitation value. We added a new sentence in line *231: "The precipitation value is obtained by multiplying the alpha and beta parameters."*

**Line 238: How were the missing data filled? Were they filled with zeros?**

The process of filling missing data in a grid point involved taking the average of the value from neighbouring grid points at the same time step. In cases where the surrounding grid points also had missing data, we resorted to using a domain-wide average for the replacement.

We added the following sentence to the manuscript in line 243: *"Gridpoints with missing data in the CMIP6 ESGCMs were filled with an average of the surrounding gridpoints. If the surrounding gridpoints had missing data as well, a domain average was applied. Afterwards, the dataset was standardized (with the same parameters used for ERA5)."*

**Line 276: How about changing "members" to "models", as the members are also used to describe 4 DL architectures (Line 384)? Similarly, in Line 48, 7 members -> 7 models.**

Where appropriate, we replaced "members" with "models".

**Line 283: Why was the base period chosen as 1981-2010 and not 1979-2014? The projected temperature increase depends on the reference period, which should be mentioned clearly in the corresponding section.**

1981-2010 was chosen because it represents a 30-year climatology that is commonly used as reference for the historical period in climate studies, and also as a climatological normal period for several institutions, such as the Portuguese and Spanish meteorological institutes. Also, this way, the base period and the future periods (except 2015-2040) consider the same 30-year length.

We added the following sentence in line 299: *"It should be noted that the projected temperature increase depends on the chosen historical period."*

**Fig. 2 and subsequent figures: I am not sure how the error bar of each boxplot is calculated. Is it related to the uncertainty of parameters in the DL model?**

In Figures 2-5, the bar in each boxplot represents the value of all gridpoints of the output of each CNN method forced with ERA5. In Figures 6-9, the bar in each boxplot represents all gridpoints of the output of all CNN methods pooled together forced with each GCM.

For clarification, we added a description of the boxplot in each figure legend.

**The error bar in Fig. 10 and the following figures seem to have different meanings compared with the previous figures, I guess. Are these related to the spread from 4 CNN methods and 7 models?**

The bar in each boxplot represents the value of all gridpoints of the output of all CNN methods forced by all GCMs, all pooled together.

As in the previous comment, we also added a description of the boxplot in each figure legend for clarification purposes.

---

## Author Comment (AC4)

**Soares et al. build on previous work on CNN downscaling to compare the performance of 4 different deep learning architectures on the Iberian Peninsula's temperature and precipitation. The study is unique in using the high-resolution observational product for Iberia and in its findings that the predictive skill was not sensitive to architecture complexity. Overall the paper was clear. Some aspects of the methodology were hard to keep track of. My comments are mostly minor.**

We would like to thank the reviewer for their time and effort in reviewing our manuscript and for providing positive and constructive feedback, and we believe that their comments helped to improve our article.

**Comments**

**A schematic would be helpful to show the main aspects of the methodology in terms of datasets used to train the model and datasets which are downscaled with the model.**

R: We added the following scheme describing the datasets used in each step of the sequence detailed in the Methodology section as the new Figure 2. The remaining figures' numbering was updated accordingly throughout the manuscript.

[Figure]

*Figure 2. Summary of the two phases of the methodology (detailed in section 2.6), describing the predictors and training and projections periods considered in each phase.*

**Do you use the same DL models that were trained on ERA5 and Iberia01 on ESGCMs or do you generate 28 unique DL models that represent the relationships between**

**each ESGCM and Iberia01? If the former, can you discuss the assumption that the relationship between coarse and fine scales is the same between ERA5 and Iberia01 as it is between ESGCM and Iberia01 (maybe in the discussion section)? I think your figures show that this assumption is valid enough. If the latter, I couldn't find that detail in the text.**

R: The same DL models trained with ERA5 data using Iberia01 were used to generate the historical and future projections of each ESGCM.

The bias-correction was applied to the ESGCMs to reduce the systematic biases of each individual ESGCM. In this way, the ESGCMs predictor values would better agree with those from ERA5. Thus, after the bias correction, we are able to assume that the relationship between each ESGCM and Iberia01 is more comparable to ERA5 and Iberia01's. Accordingly, we added the following text to the Discussion and Conclusions section, at the end of the second paragraph (line 630): "It should be noted that the DL models trained with predictors from ERA5 were used to generate the ESGCMs output for the historical and future periods. The bias correction procedure applied to the ESGCMs' predictors is an important asset that may allow their values to better agree with those from ERA5. As a result, we believe that the relationship between the predictors of both ERA5 and ESGCMs and Iberia 01 are comparable."

**I see from the comments that the DL data is to be added; are the parameters provided as well as the output?**

R: Yes, the trainable parameters were included in the published data in Zenodo (in Part 9: https://zenodo.org/records/8340297).

**L240 "The ESGCMs were bias corrected in relation to ERA5 through a simple mean-variance scaling method." How much does this affect the work on extremes, i.e., how does the variance differ?**

R: A bias correction was applied to each ESGCM to reduce their systematic biases, by adjusting their variance towards the corresponding ERA5 one (please refer to the answer to the second comment). This process allowed a better depiction of the observed climate variability by the ESGCMs, including the representation of extreme values. We assume that a better representation of historical climate variability may result in an improved depiction of future ones as well.

**L270 Is it reasonable to assume that these DL models trained on the full period would have similar attributes/predictive skill as the ones trained on shorter period?**

R: It is a reasonable assumption, especially considering the results of the performance metrics in the test period (2010-2014), shown in Figs. 3-6. By including the test period in the full training period (1979-2014), nevertheless, the loss computed during the training phase converged to similar values.

**L556 I'm not sure why you call this the climate change signal since this appears to be just the anomaly relative to some historical baseline. Are you making the case that the average of 7 members is enough to remove internal variability and give only climate signal? If so, that should be stated.**

R: We thank the reviewer for this remark, which we considered to be quite relevant. We replaced "climate change signal" with "change signal" in each legend of Figures 11-17.

**L565 It's difficult for me to make out the purple dots in contrast with the background. These figures need to be provided at higher dpi.**

R: The figures were generated with higher dpi and added to the manuscript. We also changed the color from purple to red in Figure 17 to present a better contrast with the colormap.

**L565 Can you address whether the DL architecture is producing excessively noisy results? I don't know what to make of this figure.**

R: Figure 17 represents the projected changes in extreme precipitation events. It is worth mentioning that we inspected all the computations and, in fact, a bug was disclosed linked to Figure 17. Now, we updated the figure but the noisy behavior still persists, although with a significant reduction in magnitude. Precipitation presents a highly heterogeneous spatial distribution, especially when dealing with extreme events. Often, a single extreme precipitation event can produce a considerable portion of the yearly accumulated precipitation in a given region. Additionally, the predictor-predictand relationships for extreme precipitation events are not linear. All these circumstances prevent a stable behavior of the future DL CMIP6 ESGCM projections, which may explain the spatial heterogeneities seen in Figure 17. It is also worth noticing that the overall performance of the DL models was lower in simulating precipitation than temperature. At the same time, DL produces results with higher resolution than the original ESGCM output, which could better capture local-scale extreme precipitation phenomena. Undoubtedly, this noisy behavior for DL-simulated extreme precipitation needs much more investigation, but we believe this is out of scope in this study.

**Minor comments**

**L20 Can you explain what you mean by "overall agreement"?**

R: We improved the sentence: *"(...) the DL downscaled projections demonstrate overall agreement with the CMIP6 ESGCM ensemble in magnitude for temperature projections and signal for both temperature and precipitation projections"*.

**L30 "The extensive results presented are" awkward phrasing**

R: We replaced *"The extensive results presented are (...)"* to *"The results in the report are (...)"*.

**L34 "projects worrying changes in what concerts" awkward phrasing**

R: We removed "what concerts" from the previous sentence: *"The IPCC report projects worrying changes in global-scale extreme events (...)"*.

**L36 "view of those changes being especially intense" awkward phrasing**

R: We rewrote the sentence as: *"(...) the AR6 showed particularly pronounced changes on a regional level (...)"*.

**L39 "Such disadvantage fosters" awkward phrasing**

R: We rephrased it as *"This disadvantage highlights the necessity for downscaling methods (...)"*.

**L61 unclear what "it" refers to**

R: We rewrote the sentence as *"Additionally, since SDMs are not computationally demanding, the need for large computational infrastructures is avoided"*.

**L64 "promising ones" >> "promising"**

R: We removed "ones" from the sentence.

**L64 delete " which, in turn, is a subdomain of AI" this feels like unnecessary context**

R: Removed accordingly.

**L65 "In ML...automatically" This doesn't really make sense to me. There is a training algorithm, and training data, so the models aren't training by themselves.**

R: The reviewer is correct. We rephrased the sentence as follows: *"In ML, the models learn the optimal value of their parameters automatically"*.

**L73 "an artificial analogous to" >> "an artificial analog to"**

R: Changed accordingly.

**L79 "there have been early" >> "Early"**

R: Changed accordingly.

**L80 Delete "but the results" and "enough"**

R: Changed accordingly.

**L88 "reduces for precipitation" reduces what?**

R: We added "it" after the word "reduces", referring to the uncertainty of the climate change signal mentioned at the beginning of the sentence.

**L89 "are perceived with precaution" awkward phrasing**

R: We replaced "are perceived with precaution" with "are viewed with caution".

**L91 "There have been attempts…" false parallelism here**

R: We rewrote the sentence as follows: *"There have been attempts to improve the understanding of models' reasoning (e.g., Carter et al., 2018), building an overall framework for DL studies in Earth Sciences, including weather/climate modelling and postprocessing, and generating consistent intercomparable studies (…)".*

**L95 "DL also presents…" Break this into several sentences**

R: We divided the sentence into three concise sentences: *"DL presents other general limitations, including the need for hardware (GPUs accelerate the model training while the more common CPUs can be computationally costly; Chantry et al., 2021a). Other DL limitations concern the climate research field. For example, lack of explicit physics in the DL models, and the need to split the data in a way that includes long-term patterns and trends (e.g. ENSO and global warming) in both training and test phases for long-term datasets (Schultz et al., 2021)."*

**L103 "to strong impacts on the" >> "to high"**

R: Changed accordingly.

**L104 "stronger" relative to other seasons or relative to other regions?**

R: Stronger relative to other seasons. We changed the sentence to *"Future projections point to a warming trend stronger for daytime values during summer and autumn than in other seasons, (...)"*.

**L106 "Also, it is projected a significant reduction" make active voice. "Along" >> "throughout"**

R: We rephrased the sentence as follows: *"Also, a significant reduction in mean precipitation is projected throughout (...)"*.

**L107 "aligned" >> "Concomitant"**

R: Changed accordingly.

**L121 Delete "and tested", which is unclear at this point in the paper**

R: We removed "and tested" from the sentence.

**L131 "liability" >> "shortcomings" or similar. Unclear what "liability" means in this context**

R: We corrected the sentence by replacing "liability" with "reliability".

**L161 "understandably"?**

R: Please see answer below.

**L162 "what concerns to predictors data" awkward phrasing**

R: We rewrote the sentence as: *"...understandably, the list is additionally constrained by the availability of the predictors data."*.

**L230 We need an additional column for the activation function type to make clear the difference between BMlinear and BM1**

R: We thank the reviewer for the suggestion. An additional column detailing the activation function was added to the table.

**L236 "was" >> "were"**

R: Changed accordingly.

**L271 "downscale the CMIP6 ESGCMs" here it is ambiguous whether you downscale them together or individually**

R: We specified in the revised sentence: "downscale the individual CMIP6 ESGCMs".

**L306 "slightly improvements" >> "slight improvements" Compared with Iberia01?**

R: We replaced "slightly" with "slight". The metrics were computed with Iberia01 as reference, but here we discuss and compare the DL results with the interpolated ERA5.

**L308 "(3) the four..." you mean they don't quite capture extremes, right? There's probably a more straightforward way to say this**

R: We altered the sentence as follows: *"(3) the four architectures present small biases for extreme values"*.

**L440 I find the labeling of these plots confusing because only the colored bars are called by the ESGCM's name but it seems like the grey bars also are representing those individual ESGCM outputs**

R: We replaced the grey bars representing the 1° output of each individual ESGCM with the corresponding color of each ESGCM in Figures 6 to 16.

**L448 "7 members" here it is unclear that these are referring to different CMIP6 models.**

R: The first reviewer also addressed this issue and we have replaced "7 members" with "7 models" accordingly.

**L591 "Therefore, we opted to consider..." this feels more like a methods statement**

R: We removed the sentence and specified which DL models were used in the downscaling in Line 293: *"Downscaling using the four DL algorithms is performed for each ESGCM considered (...)"*.

**L647 In what sense are they consistent? Consistent with one another in terms of 2/3 agreement?**

R: In terms of the 2/3 agreement and with previous studies. We changed the sentence as follows: *"(...) and produce high-resolution scenario-based projections, consistent with each other and with previous studies, by the use of (coarse) GCM forcing and a high-resolution training database"*.